# Aptamer-Based Probes for Cancer Diagnostics and Treatment

**DOI:** 10.3390/life12111937

**Published:** 2022-11-21

**Authors:** Xueqi Hu, Dongdong Zhang, Zheng Zeng, Linjie Huang, Xiahui Lin, Shanni Hong

**Affiliations:** Department of Medical Imaging Technology, School of Medical Imaging, Fujian Medical University, Fuzhou 350122, China

**Keywords:** aptamers, cancer diagnosis, cancer therapy, cell membrane biomarkers, extracellular biomarkers, intracellular biomarkers

## Abstract

Aptamers are single-stranded DNA or RNA oligomers that have the ability to generate unique and diverse tertiary structures that bind to cognate molecules with high specificity. In recent years, aptamer researches have witnessed a huge surge, owing to its unique properties, such as high specificity and binding affinity, low immunogenicity and toxicity, and simplicity of synthesis with negligible batch-to-batch variation. Aptamers may bind to targets, such as various cancer biomarkers, making them applicable for a wide range of cancer diagnosis and treatment. In cancer diagnostic applications, aptamers are used as molecular probes instead of antibodies. They have the potential to detect various cancer-associated biomarkers. For cancer therapeutic purposes, aptamers can serve as therapeutic or delivery agents. The chemical stabilization and modification strategies for aptamers may expand their serum half-life and shelf life. However, aptamer-based probes for cancer diagnosis and therapy still face several challenges for successful clinical translation. A deeper understanding of nucleic acid chemistry, tissue distribution, and pharmacokinetics is required in the development of aptamer-based probes. This review summarizes their application in cancer diagnostics and treatments based on different localization of target biomarkers, as well as current challenges and future prospects.

## 1. Introduction

The fast expansion of new tools for molecular diagnostics and tumor-targeted therapy has raised demand for highly specific targeting ligands for biomarkers whose expression differ in tumor cells or tissues. Owing to the intrinsic heterogeneity of human tumors, targeting ligands can aid in the identification of tumor-specific biomarkers, resulting in enhanced diagnostics, predicted therapeutic response, and as a consequence, a decrease in unnecessary therapies for unresponsive oncological patients [1,2]. The common therapeutic options today are endocrine therapy, chemotherapy and targeted therapy utilizing antibodies that recognize intracellular or extracellular cancer biomarkers. Nevertheless, they are not always effective and may result in adverse effects and drug tolerance [3]. Hence, there is an ongoing need to develop novel and effective methods for cancer diagnosis and treatment.

Aptamers provide a very interesting substitute for previous widely utilized bioaffinity materials for diagnosis or therapeutic tools for drug delivery [4,5,6,7,8,9,10]. Aptamers consist of functional single-stranded DNA or RNA that can specifically bind to targets with high affinity. The synthesis of aptamers is based on an in vitro evolution process named “Systematic Evolution of Ligands by EXponential enrichment” (SELEX) [11,12]. This technology entails iterative screening of high-affinity nucleic acid ligands for a diverse range of targets, which includes small molecules, proteins, peptides, toxins, cells and tissues. Since its introduction in 1990, approximately 20 variants of original SELEX techniques have been developed [13,14]. The basic concept of the selection procedure is similar for both DNA and RNA oligonucleotides, and it consists of three crucial steps: (i) incubating the random oligonucleotide library with the selected targets, (ii) segmentation or isolation of bound oligonucleotides from library, and (iii) recovery and amplification of the target binding candidate for use in the next cycle [15]. Therefore, sequences with desired characteristics and especially tunable specificity and affinity can be amplified selectively from the initial pool of oligonucleotides.

Compared with traditional antibodies, aptamers have the advantages of higher stability over a wide range of temperatures, pH, and solvents, making them widely used. Furthermore, aptamers are easily modifiable with various functional moieties, and can be synthesized via chemical or enzymatic approaches, earning them the moniker of “chemical antibodies”. In addition, aptamers would not cause unnecessary immunological responses due to the absence of fragment crystallizable (Fc) regions, which would bind with Fc receptors expressed on the surface of immune cells. These properties give aptamers a low immunogenicity and good biocompatibility. Moreover, specific targeting of cancer biomarkers plays a vital role in aptamers-based probes. The affinity between aptamers and cancer biomarkers become the key of diagnosis, prognosis monitoring, and targeted therapy. Therefore, a variety of aptamers optimized for targeting cancers have been selected and applied, opening up a new way for developing personalized medicines. Based on these advantages, aptamers can be ideal candidates for in vivo biosensing, which is considered to be a simple and fast way to monitor cancer diagnosis, prognosis and therapeutic response. At the same time, aptamers can be used as carriers to deliver cargo to cancer cells or tissues, so they can also be used in cancer therapy. Numerous examples are currently being tested in clinical trials, such as Pegaptanib, a treatment for age-related macular degeneration (AMD), has been shown to be effective on the commercial market [16]. Therefore, in the past decade, a variety of aptamers for cancer biomarkers have been continuously optimized and developed, opening up new avenues for the development of personalized medicines.

As we know, cancer biomarkers are present in tumor tissues or serum that include DNA, mRNA, enzyme, metabolites, transcription factors, and cell surface receptor. In the past few decades, significant and substantial progress has been made in this field. Various promising techniques based on the specific recognition of intracellular biomarkers or biomarkers on the cancer cell surface have been developed. In this review, we summarize the development and various applications of aptamer-based probes in cancer diagnostics and therapeutics according to the different location of targeted cancer biomarkers, as well as discuss current issues and future prospects in this field (Figure 1).

## 2. Aptamer-Based Probes for Cancer Diagnosis

The early diagnosis of cancer is critical for the effective treatment of the disease, and the most commonly used diagnostic method is cancer biomarker testing in serum (such as enzyme-linked immunosorbent assays, ELISA). Nevertheless, complex serum and plasma components make it difficult for detecting low-concentration cancer biomarkers, which hinders early cancer diagnosis. Some procedures are even effort-consuming and time-consuming. Therefore, owing to the high sensitivity and selectivity against low concentration targets, aptamers have become a novel and promising tool for cancer diagnosis. Unlike antibodies, the conformation of aptamers would change when they bind to their targets. This conformational change offers the opportunity to construct the unique, switchable aptamers-based probes, which cannot be realized by antibodies. Hence, aptamers have been used in the fabrication of aptamers-based probes for cancer-related biomarkers as the recognition unit and signal switch unit. The capability of aptamers to detect small differences between proteins with similar structure enables them to distinguish cancer from noncancer cells based on their distinct cell-surface homology. Thus, a variety of cancer metabolites, differentiating cells, and molecules that promote tumor growth or cancer biomarkers can be recognized by aptamers. Herein, we summarized the aptamer-based probes for cancer diagnosis according to different locations of cancer biomarkers.

### 2.1. Cancer Surface Membrane Biomarkers

The cell surface membrane (PM) plays an important role in cell structure as a physical barrier that surrounds the cell and maintains the crucial boundaries between cytoplasm and the extracellular environment. The majority of PM mass is made up of proteins with critical specific functions. These proteins determine the methods of interaction between cells and its environment, including sending and receiving chemical signals, transportation metabolites, ions, or larger molecules, attached to neighboring cells and the extracellular matrix, etc. [17,18]. The mutation, deletion and overexpression of PM proteins are related to various pathological states of cancers. Currently, PM proteins have become the target for more than half of the approved drugs [19]. Some PM proteins (such as the human epidermal growth factor receptors 2 (HER2), mucin 1 (MUC1), and epithelial cell adhesion molecules) have been identified as crucial cancer biomarkers. Hence, a large number of specific aptamers have been developed against cell surface membrane biomarkers for basic research in physiology and biochemistry and also for diagnosis, monitoring and treatment of cancers. In this review, we subdivided the aptamers for PM proteins based their biological functions by a slightly adapted classification.

#### 2.1.1. Receptors

Tyrosine kinase receptors (TKRs) are type I transmembrane receptors that become activated through binding of extracellular domain with its ligand. Following its activation or phosphorylation, the intracellular domain initiates a number of pathways, ultimately culminating in a specific response. A large number of TKRs are upregulated in malignant tumor tissues, and they are essential for tumor growth, proliferation, angiogenesis, and metastasis in cancer biology [20]. Epidermal growth factor receptor (EGFR) family of TKRs is a major player in a variety of epithelial malignancies, such as non-small cell lung cancer, breast, ovarian, colorectal and gastric cancer, and glioblastoma [21]. Aptamers for EGFR are widely developed for diagnostic strategies in related cancers. Ilkhani et al. reported an electrochemical aptamer/antibody-based sandwich immunosensor with detection limit of 50 pg/mL [22]. To overcome the limited effectiveness, Wang et al. fabricated an electrochemical paper-based aptasensor for label-free detection of EGFR by employing anti-EGFR aptamers as the recognition moiety with a sensitive detection limit of 5 pg/mL [23]. Not only that, Cheng et al. published an ^18^F-labeled RNA aptamer for PET imaging of EGFR expression in xenograft models [24]. The results showed a promising EGFR targeting molecular imaging probe for future clinical translation. Vascular endothelial growth factor receptor VEGFR2, a receptor for vascular endothelial growth factor (VEGF), is primarily expressed by angiogenic endothelial cells in the tumor stroma. Since almost all tumors depend on these cells for the transportation of oxygen and nutrients, VEGFR is regarded as a universal target for a wide variety of tumor types [25,26]. Thus, Kim et al. developed an aptamer modified magnetic nanocrystal for specific detection of VEGFR2, which enabled precise recognition of angiogenic vasculature of glioblastoma via magnetic resonance (MR) imaging [27]. In addition, human epidermal growth receptor HER2 is also a member from the EGFR family of TKRs. This protein is usually expressed in the fetal period, and its overexpression in adulthood may be associated with cancer, making it an important target for drug selection in breast cancer targeted therapy. Zhang et al. developed aptamer-based fluorescent sensor for HER2 detection [28]. In this strategy, the G-rich sequences were kept away from the dark silver nanoclusters (AgNCs) after the formation of double strand DNA (dsDNA) strands, resulting in a weak fluorescence intensity system. In the presence of HER2, the fluorescence intensity would be enhanced by the specific binding between aptamer and HER2, which caused the separation of aptamer from dsDNA and the proximity of G-bases to AgNCs. The detection limit of this sensor was 0.0904 fM. In another study, Bao et al. developed a hybridization chain reaction (HCR)-based liquid chromatography tandem mass spectrometry (LC-MS/MS) method to amplify the HER2 protein signal [29]. The aptamer of HER2 was used to trigger HCR to undergo a cascade of hybridization events, resulting in nicked duplexes resembling alternating copolymers. This design provides an enzyme-free method for detecting HER2. In most case, HER2 protein does not work alone as an individual entity, but forming dimers such as HER2:HER3 heterodimer. The presence of this heterodimer may affect the diagnosis of breast cancer [30]. Xu et al. performed co-localized fluorescence imaging of HER2:HER2 homodimers and HER2:HER3 heterodimers by using two DNA/AgNCs sequences [31]. In the presence of homodimers, the HER2-specific aptamer (Apt2) and dark AgNCs will be illuminated to produce red fluorescence; in the presence of heterodimers, the HER3-specific aptamer (Apt3) of dark AgNCs will produce green fluorescence. Compared to traditional methods, this design enables multiplex detection, providing a convenient method for multiplex imaging.

In addition, human epidermal growth receptor HER2 is a member from the EGFR family of TKRs. This protein is usually expressed in the fetal period, and its overexpression in adulthood may be associated with cancer, making it an important target for drug selection in breast cancer targeted therapy. Zhang et al. developed an aptamer-based fluorescent sensor for HER2 detection [28]. In this strategy, the G-rich sequences were kept away from the dark silver nanoclusters (AgNCs) after the formation of double strand DNA (dsDNA), resulting in a weak fluorescence intensity of system. In the presence of HER2, the fluorescence intensity would be enhanced by the specific binding between aptamer and HER2, which caused the separate of aptamer from dsDNA and proximity of G-bases to AgNCs. The detection limit of this sensor was 0.0904 fM. In another study, Bao et al. developed a hybridization chain reaction (HCR)-based liquid chromatography tandem mass spectrometry (LC-MS/MS) method to amplify the HER2 protein signal [29]. The aptamer of HER2 was used to trigger HCR to undergo a cascade of hybridization events, resulting in nicked duplexes resembling alternating copolymers. This design provides an enzyme-free method for detecting HER2. In most case, HER2 protein does not work alone as individual entity, but forming dimers like HER2:HER3 heterodimer. The presence of this heterodimer may affect the diagnosis of breast cancer [30]. Xu et al. performed co-localized fluorescence imaging of HER2:HER2 homodimers and HER2:HER3 heterodimers by using two DNA/AgNCs sequences [31]. In the presence of homodimers, the HER2-specific aptamer (Apt2) and dark AgNCs will be illuminated to produce red fluorescence; in the presence of heterodimers, the HER3-specific aptamer (Apt3) of dark AgNCs will produce green fluorescence. Compared to traditional methods, this design enables multiplex detection, providing a convenient method for multiplex imaging.

Protein tyrosine kinase 7 (PTK7), another membrane receptor protein, is a catalytically inactive receptor tyrosine kinase that plays an important role in carcinogenesis. PTK7 has been shown to be overexpressed in a variety of cancers, including gastric, colon, esophageal, lung, prostate and breast cancer [32]. Jacobson et al. developed radiolabeling of PTK7 aptamer Sgc8 with positron emission tomography (PET) isotope ^18^F via click chemistry reaction and quantified PTK7 expression in different mouse tumor models [33]. Li et al. designed an electrochemical-based method for PTK7 detection by the use of positively charged gold nanoparticles (AuNPs) and negatively charged graphene oxide (GO) and Sgc8 aptamer-based toehold-mediated strand displacement amplification [34]. This enzyme-free electrochemical sensor realized the target recycling amplification effectively with a detection limit down to 1.8 fM and applied to the detection of PTK7 in cellular debris. The cluster of differentiation (CD) proteins are cell-surface receptors that are involved in cellular functions such as activation, adhesion, and inhibition. These ubiquitous receptors express high level of CD on cells, which can be used as key biomarker in a variety of malignancies. So far, multiple aptamers have been selected that specifically target CD proteins. In 1998, the first aptamers for rat CD4 was reported by Kraus and co-workers [35]. Then, Davis et al. selected the first RNA aptamers for human CD4 molecules [36]. They applied these fluorescent labeled aptamers for staining CD4^+^ T cell in human peripheral blood mononuclear cells (PBMCs). Zhou et al. developed a CD4 RNA aptamer-functionalized glass surface to capture and enrich CD4-expressing cells [37]. In another study, Zhang et al. compared the applicability of CD4 aptamers with the CD4 antibodies for multicolored cell phenotyping in patients’ pleural fluids [38]. They simultaneously incubated the CD4 aptamers, CD4 antibodies, and CD8 and CD45 antibodies with cells from patients’ pleural fluids and were then analyzed by flow cytometric. CD30 belongs to the tumor necrosis factor receptor superfamily whose abnormal expression is closely related to Hodgkin lymphoma. Zeng et al. conjugated a CD30-specific 39-mer RNA aptamer sequence to the near-infrared dye IRD800CW reporter for imaging xenografted lymphoma tumors [39]. The results show that oligonucleotide aptamer probes can provide tumor-type-specific imaging with high sensitivity, indicating that they have certain clinical potential. Peng et al. labeled the CD30-targeting RNA aptamer with a fluorescent group and observed by flow cytometry and microscopy that the RNA aptamer showed good specific and sensitive binding to CD30-expressing lymphocytes with a detection limit of 0.3 nM [40]. CD63 (also known as lysosome-associated membrane glycoprotein, LAMP3), as the general tetraspanin protein, has been identified as the exosomal ‘star marker’ and overexpressed on the surface of exosomes from cancers. Owing to this feature, a large number of CD63 aptamer-based sensors were developed for the identification and quantification of typical exosomes. For example, Wang et al. designed a colorimetric aptasensor using graphitic carbon nitride nanosheets. The CD63 aptamer were adsorbed onto the nanosheets surface, enhancing the peroxidase activity, resulting the oxidation of tetramethylbenzidine by H_2_O_2_ and showing an intense blue color [41]. Xia et al. applied other novel nanozyme of single-wall carbon nanotube (s-SWCNT) for CD63 overexpressed exosomes with the detection limit of 5.2 × 10^5^ particles/µL [42]. In another research, Qiao et al. developed an electrochemiluminescent (ECL) aptasensor for amplified detection of CD63 overexpressed exosomes in serum samples [43]. The recognition of CD63 to this aptasensor allowed for the forming of a G-quadruplex/hemin DNAzyme to catalyze the decomposition of H_2_O_2_, leading to the signal change of ECL with the detection limit of 7.41 × 10^4^ particles/mL. Wang et al. reported a novel strategy for sensitive detection of CD63 overexpressed exosomes by surface plasmon resonance (SPR) with dual gold nanoparticle (AuNP)-assisted signal amplification and achieved a limit of detection down to 5 × 10^3^ particles/mL [44]. Interestingly, Zhang et al. developed a novel CD63 aptamer-based fluorescence polarization assay for separation-free, amplification-free and sensitive quantification of exosomes in human plasma directly [45]. The detection limit of this assay was 500 particles/µL and the total assay time was approximately 30 min with just one mix-and-read step to achieve high sensitivity. (Examples are listed in Table 1).

#### 2.1.2. Cell Adhesion Molecules

Cell adhesion molecules (CAMs) are a huge and diverse group of membrane-bound proteins known as morphoregulatory molecules that have an impact on cellular functions. The definition shows that, contrary to what their names may imply, these proteins are involved in processes other than cell–cell or cell–matrix adhesion [46]. Epithelial cell adhesion molecule (EpCAM), a typical CAM, is highly associated with tumor proliferation and overexpressed in the majority of solid tumors, while being expressed at low levels in a variety of human epithelial tissues [47]. Shi et al. developed a fluorescence resonance energy transfer (FRET) biosensor on the basis of graphene quantum dots-aptamer and MoS_2_ nanosheets for the detection of EpCAM with the detection limit of 450 pM [48]. To enhance sensitivity, Chen and co-workers constructed an ultrasensitive and enzyme-free fluorescent method for EpCAM detection based on the toehold-aided DNA amplification strategy with the detection limit of 0.1 ng/mL [49]. Besides the quantification of EpCAM, Heo et al. developed the maleimidyl magnetic nanoplatform based on the conjugation of EpCAM aptamer and magnetic nanomaterials for magnetic resonance imaging (MRI) of gastric cancer [50]. Besides, EpCAM has also been identified as a general exosomal biomarker in colorectal, ovarian and pancreatic cancer [51,52,53,54]. Wang et al. designed an amplified aptasensor using graphene oxide and fluorescent labeled aptamers for CD63 and EpCAM on colorectal cancer exosome with a detection limit of 2.1 × 10^4^ particles/µL and can be applied for clinical colorectal cancer serum samples in 5 µL [51]. Zhang et al. reported an ECL biosensor by using EpCAM aptamer modified two-dimensional material Ti_3_C_2_ MXenes nanosheets for exosomes quantification in clinical breast cancer serum sample with the detection limit down to 125 particles/µL [53]. To further improve the detection limit, Zhao et al. constructed a target-triggered 3D DNA walking machine and exonuclease III (Exo III)-assisted electrochemical ratiometric biosensor for amplified detection of exosomes with detection limit of 13 particles/µL [55]. To apply the aptamer-based diagnosis in circulating tumor cells (CTCs), Hashkavayi et al. developed an aptamer-based gold nanostar structure for detection of EpCAM overexpressed-CTCs using a dual signal amplification strategy based on rolling circle amplification (RCA) coupled with the hemin/G-quadruplex complex [56]. With this system, target cancer cells could be analyzed in the range of 5 to 10^7^ cells/mL with a detection limit down to 1 cell/mL.

Another important family of CAMs are the integrins. Due to their interactions with other cells and with the extracellular matrix, integrins are composed of heterodimer receptors that are essential in the control of the cell cycle, cellular shape, and motility. In particular, integrins containing the alpha subunit are prodigious overexpressed in tumors, such as ανβ3, ανβ5, ανβ6, α5β1, etc. Nevertheless, there are few applications about integrins aptamers-based cancer diagnosis. Lim et al. developed aptamer-conjugated magnetic nanoparticles to realize precise detection of integrin ανβ3 expressing cancer cells by using magnetic resonance imaging [57]. Fechter et al. selected a novel aptamer to target integrin α5β1-expressing cells, and applied it to distinguish glioblastoma cell lines and tissues from patient-derived tumor xenografts according to their α5 expression levels [58]. (Examples are listed in Table 2).

#### 2.1.3. Cell Membrane-Associated Enzymes

The membrane-associated enzymes have also become the ideal tumor biomarkers, which are mainly involved in the maintenance of cellular functions, uptake/secretion/proteolysis of proteins, and extracellular matrix remodeling. They control cell–cell and cell–matrix interactions, maintaining the homeostasis of the cell. Prostate-specific membrane antigen (PSMA), as a type II membrane-bound peptidase, found mainly in prostate tissues [59], and it is regarded as a functional cancer biomarker because it is abundantly upregulated on prostate carcinoma cells and on the neovasculature of most other solid tumors [60]. Lupold et al. reported the first RNA aptamers that target PSMA. These aptamers were the first reported RNA aptamers selected to bind tumor-associated membrane antigen as well as the first application of RNA aptamers to prostate cancer cell lines with a low detection limit of 2.1 nM [61]. These aptamers were modified to AuNPs by Walter et al., who then utilized them in tissue microarrays to detect PSMA in human prostate cancer tissue [62]. As for the imaging diagnosis, Fan et al. achieved the targeted diagnosis of PSMA under ultrasound imaging by coupling PSMA aptamers to lipid nanobubbles [63]. To overcome the poor visuality and veracity of ultrasound imaging for diagnosis, Gu et al. developed a nanoultrasound contrast agent by modifying multi-walled carbon nanotubes (MWCNTs) with polyethylene glycol (PEG) and anti-PSMA aptamers [64]. This strategy showed their good ability to quantify PSMA both in vitro and in vivo. Based on the expression of PSMA, prostate cancer has two types of cells, PSMA (+) and PSMA (−) cells [59,65]. Therefore, Min et al. developed an RNA/peptide dual-aptamer probe based electrochemical detection method to simultaneously detect PSMA (+) and PSMA (−) prostate cancer cells [66].

The matrix metalloproteinases, also known as MMPs, are another most prominent invasion-associated proteases that have been found to be overexpressed in various types of cancers [67]. Several aptamer-based probes have been developed using upregulated membrane-bound or membrane-associated proteolytic enzymes, such as MMP-2, MMp-7, and MMP-9, for cancer diagnosis. Gomes et al. selected an RNA aptamer containing 2′-fluoro, pyridine ribonucleosides, that showed a high affinity for MMP9 and distinguished it from other MMPs (MMP-2 and MMP-7) [68]. Then, this aptamer labeled with ^99^mTc labeling was applied to ex vivo imaging slices of human brain tumors. To improve sensitivity, Scarano et al. reported a dual aptamer-based piezoelectric biosensor which can detect the active form of MMP-9 with very high sensitivity [69]. This method showed that the detection limit in untreated serum were 1.2 pM. Recently, Kim et al. developed a photoacoustic contrast agent that introduced programmed hybridization/dehybridization of MMP-9 aptamers on the surface of gold nanospheres (AuNSs) for ultrasound-guided photoacoustic imaging of human MMP-9 in human breast cancer [70]. (Examples are listed in Table 3).

#### 2.1.4. Other Membrane-Associated Proteins

The mucin 1 (MUC1) is a type I transmembrane protein that normally has a polar distribution. Moreover, owing to its presence in a variety of malignant tumors (such as lung, breast, ovarian, bladder carcinomas, prostate, gastric, etc.), MUC1 has been used as an important cancer biomarker [71,72,73]. Ma et al. constructed a sensitive MUC1 aptamer and MUC1 antibody-based sensor for MUC1 detection by using carbon dots with the sensitive detection limit of 2 nM [74]. In another study, a simple aptamer-based sensor for MUC1 image and quantitative analysis was developed by the combination of silver nanoclusters (AgNCs) and aptamers with the detection limit of 0.05 nM [75]. Another crucial diagnosis technique is aptamer-based nucleic acid chain amplification, which is highly sensitive against targets. Peng et al. constructed a simple and effective signal amplification method without any labels or enzymes for aptamer-based sensitive detection of MUC1 with a detection limit of 2.8 nM [76]. MUC1 aptamers were also applied in the detection of CTCs. Wang et al. developed a dual-aptamer (VEGF and MUC1 aptamers) nanoparticle-mediated signal amplification strategy for cancer cell colorimetric detection with a detection limit as low as 10 cells/mL [77]. In other research, Cao et al. developed electrochemical aptasensor using MUC-1 aptamer-bound carbon nanospheres with superb analytical sensitivity for colon CTCs (detection limit of 40 cells/mL) [78]. Zhang et al. found MUC1 was also highly expressed on the surface of human breast cancer cell line (MCF-7 cells)-derived exosomes compared to the normal breast cell-derived exosomes [79]. In their work, an aptasensor based on MUC1 aptamer was designed and labeled with a luminophore and quenching group. The hairpin-like MUC1 aptasensor can be opened by MCF-7 exosomes and then the luminophore would be separated from the quenching group, resulting in strong fluorescence with a detection limit of 4.2 × 10^4^ particles/μL. Huang et al. developed a label-free electrochemical MUC1 aptasensor for detection of gastric cancer exosomes by combining RCA and hemin/G-quadruplex system with a detection limit of 9.54 × 10^2^ particles/mL [80].

Nucleolin (NCL) is also another particular cancer membrane-associated protein biomarker, which is highly expressed both intracellularly and on the cell surface in several cancers [81]. Li et al. realized label-free detection of nucleolin by introducing an AS1411 aptamer-based microcantilever biosensor with the detection limit of 1.0 nM [82]. Li et al. labeled 3′-NH_2_-modified AS1411 with ^64^Cu by using four different chelators and assessed them in in vitro cellular uptake by micro position emission tomography/computed tomography (microPET/CT) imaging [83]. Noaparast et al. labeled 5′-NH_2_-modified AS1411 with ^99^mTc as an ideal gamma emitter radionuclide using a hydrazinonicotinic acid (HYNIC) chelator and evaluated its stability and affinity to cancers and normal cells [84]. To realize to multimodal cancer-targeted imaging, Hwang and co-workers designed a cobalt-ferrite nanoparticle surrounded by fluorescent rhodamine within an AS1411 aptamer modified sillica shell matrix for fluorescent and magnetic resonance imaging [85]. Recently, Wu et al. published an AS1411 aptamer-based phosphorescent nanoprobe, which acts as tumor-targeting moiety and Λ-[Ru(bpy)_2_(p-EPIP)](ClO_4_)_2_ (RuPEP, bpy = bipyridyl, p-EPIP = 2-(4-ethynyl phenyl)-1H-imidazo [4,5-f] with excellent luminescent property act as phosphorescence probe to highlight breast cancer [86]. Huang et al. developed a dual-signal amplification platform for detection of leukemia cell-derived exosomes with anti-CD63 antibody modified magnetic bead conjugates (MB-CD63) and AS1411 based primer-padlock, which combined use of aptamer recognition, magnetic enrichment and rolling cycle amplification. With this strategy, as low as 100 particles/μL exosomes could be detected. To further improve sensitivity, Zheng et al. developed a dual-aptamer recognition strategy for ultrasensitive detection of exosomes from cervical cancer HeLa cells with a detection limit of 1.85 × 10^3^ particles/mL [87]. In this work, target exosomes were captured firstly by CD63 aptamer modified magnetic beads and then combined with AS1411 aptamer and myc monomer for realizing fluorescent signal amplification. In addition, the strategies for nucleolin-overexpressed cancer diagnosis by CTCs were also well-developed. Hua et al. reported a strategy based on MUC1 aptamer functionalized magnetic beads and AS1411 modified quantum dots based nano-bio-probes [88]. The probes displayed a similar optical and electrochemical performance to free CdTe QDs and high affinity remained on nucleolin-overexpressing cells with the detection limit of 201 cells/mL by photoluminescence and 85 cells/mL by the square-wave voltammetry assay. Ou et al. developed a sandwich-type cytosensor to analyze cancer cells, which was based on the metal organic framework (MOF) and a DNA tetrahedron linked aptamer (AS1411 and MUC1 aptamers) with the detection limit of 6 cells/mL [89].

Vascular endothelial growth factor (VEGF), as a signaling protein, plays a critical role in the regulation of vascular formation and permeabilization [90]. The VEGF family consists of five members: VEGF-A. VEGF-B, VEGF-C, VEGF-D, and placenta growth factor, each of which has many isoforms. For example, VEGF-A is divided into four major isoforms: VEGF121, VEGF165, VEGF189, and VEGF206. When carcinogenesis occurs, it is usually accompanied by the formation of tumor blood vessels, so the abnormal increase of VEGF is an important biomarker for cancer diagnosis [91]. Moghadam et al. developed a signal-on nanobiosensor based on bivalent aptamer-Cu nanocluster for the detection of VEGF165 with the detection limit of 12 pM [92]. To enhance the sensitivity, Li et al. introduced an amplified fluorescence sensing platform for VEGF and ATP detection in real biological samples [93]. The platform was developed based on nicking endonuclease-assisted signal amplification and the high quenching ability of GO with a low detection limit of 1 pM. In order to decrease the effect of complex environments, Zhu et al. constructed an AgNPs-enhanced time resolved fluorescence sensor for VEGF165 detection by using long-lived fluorescent Mn-doped Zinc Sulfide Quantum Dots (ZnS QDs) [94]. This strategy offered high signal-to-noise ratio in detection and avoided the disturbance of the intrinsic fluorescence of some proteins. To realize the image of VEGF165, You et al. combined superparamagnetic iron oxide (USPIO) nanoparticles with VEGF165 aptamers by chemical synthesis to prepare a magnetic resonance imaging (MRI) probe for MRI imaging of VEGF165-expressing tumors in vivo [95]. (Examples are listed in Table 4).

### 2.2. Extracellular Cancer Biomarkers

The extracellular environment has a large impact on tumor and non-tumor tissues, especially in the concentrations of extracellular cancer biomarkers in body fluid [96]. Abnormal changes in cancer biomarker levels in body fluids are measuring standards of disease progression renewal. Therefore, the detection of these abnormal changes is often the key to realize early cancer diagnosis. Currently, the efforts are focused on liquid biopsy that relies on the presence of specific biomarkers in the body fluid of cancer patients. These cancer biomarkers are difficult to detect due to their low concentration levels in a high protein content medium. Enzyme-linked immunosorbent assays (ELISA) are the gold standard method for cancer biomarkers detection in body fluid, which rely on antibodies. Nevertheless, they suffer from some limitations, such as batch to batch variations during their production, and the challenging and cumbersome technique needed for generating specific monoclonal antibodies (especially against non-immunogenic molecules) [97,98]. For this reason, aptamers as novel receptors overcome these limitations owing to their unique characteristics of good stability, biocompatibility, safety, efficiency and non-immunogenicity. Herein, we summarized the various applications of aptamers in cancer diagnosis by introducing extracellular cancer biomarkers that have been detected in different body fluids.

Platelet-derived growth factor (PDGF), as one of serum components, has been proven to promote the proliferation of arterial smooth muscle cells [99]. The PDGF family is composed of four ligands: PDGF-A, B, C and D. PDGF-BB, a homodimer of PDGF-B, is an important cancer biomarker in diagnosis and recognition of cancers. During the past few years, aptamer-based recognition and detection of PDGF-BB for cancer diagnosis has been well-developed. Huang et al. developed an aptamer-modified AuNPs (Apt-GNPs) for the sensitive detection of PDGF-BB based on observing the changes in the color and extinction of the specific aptamer and GNPs by cause of aggregation [100]. This sensor was with a low detection limit of 35 nM and applied to protein analysis and cancer diagnosis. In another study, Tang et al. reported a strategy integrating rolling-circle amplification (RCA) and aptamer-based DNA enzyme-catalyzed colorimetric reaction for sensitive detection of PDGF-BB [101]. The PDGF-BB was recognized using primary aptamer-functionalized microbeads in a sandwich approach, and a secondary aptamer was attached to an RCA primer/circular template complex. The detection limit of this strategy was 8.2 fM. Zhang et al. constructed a AuNPs colorimetric sensor for detecting PDGF-BB by target-triggered strand displacement amplification system [102]. An obvious AuNPs color change can be observed when PDGF-BB concentration was as low as 4.0 nM. Interestingly, Ye et al. proposed a novel and simple aptamer-based one-two-three cascade DNA amplification surface-enhanced Raman scattering (SERS) strategy for the detection of PDGF-BB with the detection limit down to 0.42 pM [103]. Besides the strategies based on colorimetric and Raman scattering readout, fluorescent method is a simple and strong strategy because of its high sensitivity, rapid, simple and comparatively cost-less [2]. Taking advantages of fluorescent method, various aptamer-based fluorescent strategies have been developed for PDGF-BB detection. However, the labeling of aptamer with fluorophores and quenchers are the main limitations of the existing methods, which are time consuming and expensive. To overcome these limitations, Babu et al. reported an assay for label-free luminescent detecting PDGF by conjugating aptamer to hydrophobic Ru (II) complex as sensor system [104]. This method could detect the PDGF in a mixture of proteins, down to 0.8 pM. Wang et al. introduced a label-free and enzyme-free aptasensor for PDGF-BB quantification by using target-triggered hybridization chain reaction amplification and grapheme oxide (GO)-based selective fluorescence quenching with a detection limit of 1.25 pM [105]. Wang et al. reported another label-free fluorescent aptasensor for PDGF-BB detection by photo-induced electron transfer between DNA-AgNCs and G-quadruplex/hemin complexes. Binding of PDGF-BB to its aptamer caused a conformational change of DNA and the release of G-quadruplex sequence, which resulted in fluorescence change of the system [106]. In addition, Lin et al. developed a FRET based aptasensor using upconversion nanoparticles (UCNPs) as donor and AuNPs as acceptor for the PDGF-BB detection in blood serum of lymphoma patient with a low detection limit of 10 nM [107]. Compared to other biosensing methods, such as optical detection, the electrochemical aptasensors showed highly applicable and attractive for developing point-of-care cancer diagnosis tools owing to its advantages of disposability, accuracy, the ability to work with complex samples, easy control, rapid response, possible of usage for online control, etc. [108,109]. Recently, Jiang et al. developed a dual signal amplification for electrochemical aptasensing of PDGF-BB using hydroxyapatite nanoparticles (HAP-NPs) [110]. The phosphate group in both HAP-NPs and the aptamer reacted with molybdate to create a redox-active molybdophosphate precipitated on the surface of a glassy carbon electrode (GCE). When a voltage of 0.21 V (vs. Ag/AgCl) is applied, a current is generated whose intensity depended on the concentration of analyte. This work was applied to the determination of PDGF-BB in serum sample with a detection limit of 50 fg/mL.

Alpha fetoprotein (AFP), an albuminoid protein, is secreted from the fetal liver in the embryonic stage and usually undetectable in healthy adults [111]. Nevertheless, the increasing AFP concentration in adult serum leads to various diseases, including liver cancer. Thus, it is critical to develop rapid, ultrasensitive and selective strategies for early detection of AFP. Recently, several biosensors based on target-induced aptamer switched mode have been developed for AFP detection. There are some similar designs, such as the absorption of FAM-labeled aptamers on AuNCs [112], palladium nanoparticles (PdNPs) [113], and graphene oxide (GO) [114], resulting in fluorescence quenching, respectively. The conformation of aptamer changed in the presence of AFP, resulting in a weakening of the interaction between the fluorophore and the nanomaterials. Therefore, the fluorescence intensity of FAM could be recovered. In the above strategies, the detection limits of AFP were estimated to be 6.631 ng/mL for AuNCs [112], 1.4 ng/mL for PdNPs [111], and 0.909 pg/mL for GO [114]. Besides the labeled sensing strategies, Bao et al. developed a label-free fluorescent aptasensor based on target-induced aptamer switched mode for AFP detection [115]. This method showed a simple “mix-and-detect” procedure and was capable of rapid detection within 5 min. In another study, Zhou et al. used a sandwich binding type to construct the fluorescent aptasensor for AFP detection [116]. This aptasensor was based on FRET where the AFP aptamer labeled cadmium telluride quantum dots (CdTe QDs) as a donor and anti-AFP antibody conjugated AuNPs as an acceptor. The interaction between aptamer, target, and antibody made the QDs and AuNPs close enough in the presence of AFP, which lead to a fluorescence change with a detection limit of 400 pg/mL.

Carcinoembryonic antigen (CEA), a broad-spectrum cancer biomarker, was used to diagnose a variety of clinical disease, including pancreatic cancer, ovarian cancer, gastric cancer, colorectal cancer [117]. Specifically, the concentration of CEA in blood is significantly higher in cancer patients than that in healthy humans. Thus, the precise quantification of CEA in blood is of great significance in early diagnosis. Zhang et al. developed fluorophore-labeled CEA aptamer-absorbed molybdenum disulfide (MoS_2_) nanosheets for the analysis of CEA with a detection limit of 34 pg/mL [118]. To overcome the poor photostability of most organic fluorophore, fluorescent nanomaterials have been used for developing the aptamer-based sensor for CEA detection. Yang et al. employed AuNPs linked with CEA aptamer and its complementary DNA (cDNA), resulting in surface-enhanced fluorescence [119]. The detection limit of this strategy was 3 pg/mL in human blood sample. Owing to the advantage of near-infrared excitation and visible emission property, upconversion nanoparticles of carbon nanoparticles (CNPs) were also used for designing the aptamer-based CEA detection [120]. In another study, Shao et al. developed an aptamer-based FRET sensor between near-infrared carbon dots (NIR-CDs) and gold nanorods (AuNRs) with a low detection limit of 0.02 pg/mL [121]. In addition, Chen et al. developed a label-free detection method for CEA by applying CEA aptamer-based double stranded DNA (dsDNA) as the template for generating copper nanoparticles with the detection limit of 6.5 pg/mL [122]. In order to enhance sensitivity, many signal amplification technologies have been developed to quantify CEA. He et al. designed a smart DNA walker biosensor for label-free quantification of CEA based on exonuclease III-assisted target recycling and DNA walker amplification strategy and showed a low detection limit of 1.2 pg/mL [123]. However, this enzyme-based amplification strategy was highly reliant on enzyme activity. Therefore, an enzyme-free and label-free ratiometric aptamer-based sensor was reported for sensitive CEA detection by a recycling amplification strategy [124].

8-Hydroxy-2-deoxyguanosine (8-OHdG), a small molecule, is one of the major products of DNA oxidative stress injury. The concentration of 8-OHdG detected in human urine can indicate the level of whole-body DNA oxidative damage and has also been discovered to correlate with various cancers [125]. Therefore, the sensitive quantification of 8-OHdG is critical in clinical diagnosis of cancer. Liu and coworkers utilized 8-OHdG aptamer as a recognition probe and N-methyl mesoporphyrin IX (NMM) as a fluorescent reporter to develop a label-free and low-cost fluorescence biosensor for the detection of 8-OHdG with the detection limit of 1.19 nM [126]. In order to enhance sensitivity, Wei et al. developed a 3-dimensional DNA nanomachine to improve the determination of 8-OHdG [127]. This nanomachine was highly efficient because one aptamer can release hundreds of signal reporters, allowing the detection sensitivity to be increased and the detection limit down to 4 pM. In another research, Fan et al. developed a simple electrochemical aptasensor for sensitive detection of 8-OHdG in human serum and bladder cancer urine samples based on it triggered polyaniline deposition on tetrahedral DNA nanostructure with the detection limit of 1 pM [128]. Besides the fluorescent and electrochemical strategies, Gan et al. reported an innovative magnetic aptamer adsorbent Fe_3_O_4_-aptamer magnetic nanoparticles for the selective extraction of 8-OHdG in complex human urinary samples [129]. The extracted 8-OHdG was analyzed by high performance liquid chromatography-mass spectrometry (HPLC-MS) and the detection limit of this method was 0.01 ng/mL. (Examples are listed in Table 5).

## 3. Aptamer-Based Cancer Therapy

Conventional cancer treatments such as chemotherapy, photodynamic therapy, photothermal therapy and radiotherapy may cause serious negative effects to patients due to their associated non-specific toxicity. In order to reduce these negative effects, a personalized and targeted therapy concept has been developed. Aptamer-based targeted therapies and specific drug delivery systems are one of the most popular technologies. Aptamers-based therapy typically employs one of two strategies: (1) aptamer as therapeutic agent, such as antagonist for blocking the interaction of disease-associated targets or agonist for activating the function of target receptors; (2) aptamer as delivery agent for delivering other therapeutic agents to the target cells or tissue. These investigations demonstrated numerous advantages of aptamer technology over protein-based antibody treatments.

### 3.1. Aptamer as Therapeutic Agent

Aptamers are a group of potential therapeutic agents. They are known as “synthetic antibodies” because of their synthetic nucleic acid-based nature and excellent specificity and affinity for both protein and non-protein targets. The majority of aptamers utilized for therapeutic purposes are either chosen by in vivo selections using appropriate model systems or through in vitro selection utilizing a purified protein or receptor [130]. In clinical trials, therapeutic aptamers can be utilized as antagonists and agonists. Antagonist aptamers block or inhibit the integration of targets relevant to disease via protein–protein interaction or protein–receptor–ligand interaction [15]. Agonist aptamers can active the target receptors and also can be utilized as the carrier to carry the cargo to the target cells or tissues. Numerous studies have been reported that therapeutic aptamers applied for cancer therapy.

#### 3.1.1. Cancer Surface Membrane Biomarkers-Based Therapy

The complicated cell membrane protein fractions are often associated with carcinogenesis and metastasis, making them an ideal target for cancer therapy. Numerous therapeutic aptamers targeted for cell membrane biomarkers have been well-developed. For example, vascular epidermal growth factor VEGF is a highly specific factor that promotes the growth of vascular endothelial cells in cancers. Among them, VEGF-A can promote the formation of new blood vessels in cancers and activate vascular epidermal growth factor receptor VEGFR-1 and VEGFR-2 [131]. Yoshitomi et al. screened DNA aptamers Apt01 and Apt02 with high specific targeting to VEGFR-1 and VEGFR-2 by SELEX technique [132] (Figure 2a). The results indicated that Apt02 could form a stable G-quadruplex structure and accelerate tube formation in human umbilical vein endothelial cells faster than Apt01. Thus, Apt02 may have the potential to replace VEGF-A. Reyes-Reyes et al. demonstrated the antiproliferative activity of AS1411 in various cancer cell lines correlated with it capacity to stimulate micropinocytosis [133] (Figure 2b). From their results, AS1411 binding to nucleolin may cause Rac1 hyperactivation, leading to methuosis of target cells. Prabu et al. developed a novel RNA aptamer, SECURA-3, which specifically targeted human pituitary tumor transforming gene 1 (PTTG1) and exhibited antagonism by antagonizing the interaction between PTTG1 and chemokine receptor 2 CXC receptor2 (CXCR2) [134] (Figure 2c). In addition, programmed cell death-1/programmed cell death-ligand 1 (PD-1/PD-L1) is an important immune checkpoint, controlling the induction and maintenance of immune tolerance within the tumor microenvironment [135]. Gao et al. obtained an aptamer named PL1 that specifically binds to PD-L1 by cell-SELEX technology with the engineered PD-L1-expressing cells as a target [136] (Figure 2d). Aptamer PL1 could restore the proliferation and interferon-γ (IFN-γ) rescue from T cells suppressed by the PD-1/PD-L1 axis and inhibit the growth of colon tumor 26 (CT26). Based on the principle that blocking the PD-1/PD-L1 pathway can reduce immunosuppression and enhance anticancer effect, Li et al. further constructed a tetravalent DNA nanostructure that formed by four PD-L1 aptamers for significantly improve the antitumor efficacy in vivo [137]. In addition to single-targeting aptamers, bispecific aptamers can also act as modulators to bind both target receptors and paired proteins at the same time when they get close to cell membrane. Wang et al. designed a bispecific aptamer to target transferrin receptor protein (TfR) and tyrosine protein kinase (Met) protein inducing artificial protein pairing and inhibiting the function of target proteins by substantial steric hindrance [138] (Figure 2e). Compared with a single aptamer, the bispecific aptamer probe improved the selectivity and effectively modulates receptor function, thus providing a new way to develop novel therapeutic drugs.

Apart from the aptamer antagonists, aptamers can also act as agonists to improve cancer therapy. Only a few aptamers have been screened to act as therapeutic agonists. These include RNA aptamers targeting vascular epidermal growth factor 3 (VEGFR-3) [139], OX40 (a member of the tumor necrosis factor receptor super family, also known as CD13) [140], 4-1BB (also known as CD137) [141], CD28 [142], CD40 [143] and DNA aptamers targeting human VEGFR-2 [144] and the insulin receptor [145]. These aptamers usually target co-stimulatory receptors (CD28, CD40, and 4-1BB) of the immune signaling pathway [145]. Effective co-stimulation on the surface of antigen presenting cells or T cells induced by the interaction of a co-stimulatory receptors and their ligands plays an important role in enhancing antitumor immunity [146,147]. The first-generation agonistic RNA aptamer was screened by SELEX against the extracellular domain of 4-1BB. Compared with monomeric form, the bivalent 4-1BB aptamers were able to co-stimulate the T-cell activation, causing the tumor regression in mice models [141] (Figure 3a). Dollins et al. assembled two copies of OX40 RNA aptamers on an oligonucleotide-based scaffold, resulting in the activation of OX40 receptor on primed T cells in vitro and enhanced antitumor responses in mice [148] (Figure 3b). Pratico et al. combined two OX40 RNA aptamers containing a 5′ terminal biotin group by streptavidin linker to stimulate OX40 on human T cells and improve cell proliferation and IFN-γ production [140]. In another study, Pastor et al. isolated two RNA aptamers (CD28apt2 and CD28apt7) against murine CD28 [142] (Figure 3c). As monomer, the CD28apt2 aptamer blocked the interaction of CD28 with ligand B7.2, while the other aptamer CD28apt7 was inactive. However, the dimerization of these two aptamers can offer an artificial costimulatory signal. Soldevilla et al. isolated two 2′-fluoropyrimidine-modified RNA aptamers against the murine CD 40 receptor [143]. The CD40 aptamers were designed as a structure with three different functions. The agonistic bivalent aptamers enhanced proliferation and activation of B lymphocytes, and accelerated recovery of bone marrow aplasia, whereas the antagonistic monovalent aptamer inhibited proliferation of B cell lymphoma and improve the overall survival in mice (Figure 3d).

#### 3.1.2. Other Biomarkers-Based Therapy

Apart from cell membrane protein targeted therapeutic aptamers, there are several cancer-associated biomarkers, such as extracellular biomarkers, intracellular biomarkers, CTCs and exosomes, that can also be served as therapy target for therapeutic aptamers. Biesecker et al. obtained an aptamer specifically targeting extracellular biomarkers human complement C5 through SELEX technology [149]. In this study, they found that this aptamer can be derived to the complement C5 component and can inhibit the hemolytic activity of human serum. Besides the biomarkers from extracellular or intracellular, CTCs also play a vital role in the development of metastasis. The increasing quantity of CTCs in circulation is connected with a poor prognosis. Therefore, targeting CTCs provides a novel technique for disrupting the cancer metastatic cascade [150]. Orava et al. mixed the anti-CEA aptamer with CEA-overexpressing colonic carcinoma cells and then injected these pretreated cells into a mouse model intraperitoneally [151]. The results showed a significant reduction in number and volume of metastatic nodules. Wang et al. developed an aptamer against sialyl Lewis X (sLex), which assists cancer cells in binding endothelial-selection (E-selection) and metastasis [152]. Their results confirmed that anti-sLex aptamer can inhibit adhesion and migration of HepG2 cells.

Similarly, there is accumulating evidence that the critical role of exosomes in the development of cancer metastasis has piqued the interest of some researchers in identifying aptamers that target exosomes and developing therapeutic aptamers to treat cancer metastasis. Lately, Esposito et al. developed a novel differential SELEX technology, called Exo-SELEX, to identify aptamers that specifically bind cell-derived exosomes [153]. This technology selected novel Ex-50.T aptamers by using the most aggressive subtypes of breast cancer (triple-negative and HER2) and the results showed that Ex-50.T aptamers were the functional inhibitor of exosome cellular uptake and antagonizes cancer exosome-induced cell migration in vitro. Another approach for therapeutic aptamers to target exosomes is with the help of nanotechnology. Xie et al. have exploited a well-known fact about mesoporous silica nanoparticles being taken up by the liver and then eliminated through the small intestine to construct the nanoparticles that bind and eliminate circulating exosomes specifically from the blood [154]. The mesoporous silica nanoparticles in this assay were decorated with an aptamer that particularly targets the epidermal growth factor receptor on the surface of exosomes. The results demonstrated that these aptamer-decorated nanoparticles can inhibit the pulmonary metastasis formation in a subcutaneous murine tumor model.

### 3.2. Aptamer as Delivery Agents

Aside from their utility as stand-alone therapeutics, aptamers can also serve as chaperones for another therapeutic. A variety of cell type-specific aptamers have been coupled with therapeutic drugs (such as siRNA, microRNA, anti-miR, therapeutic aptamer, chemotherapeutics, or toxins) or delivery vehicles (such as organic or inorganic nanocarriers) for cell type-specific delivery. Owing to the high specificity and affinity of aptamers, therapeutic compounds can be targeted to the desired cells or tissues, enhancing their local concentration and therapeutic efficacy. The following will introduce the several examples of aptamers-mediated delivery methods according to different locations of targets.

#### 3.2.1. Cancer Surface Biomarkers-Targeted Delivery Therapy

Cancer membrane biomarkers are functional molecules that play significant roles in numerous biological processes, including signal transduction, cell adhesion and migration, cell-cell interactions, and communications between the intra and extra-cellular environments [155]. The tumorigenesis is often associated with abnormal expression of cell membrane biomarkers [156]. It is estimated clinically that over 60% of cancer-targeting drugs, including small molecule inhibitors and therapeutic antibodies, are thought to target cell surface biomarkers, making them appealing for disease treatment. In recent years, numerous aptamers targeting cell surface biomarkers have been widely explored for various cancer therapy. Hicke et al. created a term “escort aptamers” in 2000 and demonstrated that aptamers can be employed as delivery agent for therapeutic reagents [157]. Intracellular delivery is achieved by loading siRNA, drugs, toxins, enzymes, photodynamic molecules and radionuclides using aptamers for internalized cell surface receptors. Notably, aptamer-based targeted delivery of these drugs not only enhances the drugs’ therapeutic activity but also significantly decreases the non-specific effects.

The first nanoparticle-aptamer conjugate was generated by using the anti-PSMA A10 aptamer. Farokhzad et al. reported docetaxel (Dtxl)-encapsulated nanoparticles formulated with biocompatible and biodegradable poly(D,L-lactic-co-glycolic acid)-block-poly(ethylene glycol) (PLGA-b-PEG) copolymer and surface functionalized with anti-PSMA A10 RNA aptamers [158]. The results showed that these nanoparticles could target prostate lymph node barcinoma of the prostate (LNCaP) epithelial cells efficiently, inhibiting tumor growth in an in vivo LNCaP flank tumor model. In order to enhance the stability of RNA aptamers, Leach et al. developed a DNA-RNA hybrid aptamer by connecting a small single-stranded DNA to RNA aptamer A10-3-J1 [159]. Afterwards, the hybrid aptamer was coupled by superparamagnetic iron oxide nanoparticles (SPION) and loading the doxorubicin (DOX) to form a targeted delivery system. In another research, Chen et al. constructed a PSMA aptamer-anchored nanoparticles for systemic delivery of docetaxel (DTX) to prostate cancer and enhancing the antitumor effect in vitro and in vivo [160]. Gene therapy also plays a critical role in cancer therapy. However, the lack of targeting and safe delivery systems limits its application. To overcome these challenges, McNamara et al. designed a simple aptamer-siRNA chimera (A10-siRNA) for targeting the expression of survival genes in PSMA-expressing prostate cancer cells [161]. To further enhance the therapeutic efficiency, Zhen et al. developed PSMA aptamer-conjugated cationic liposomes to efficiently and flexibly introduce therapeutic clustered regularly interspaced short palindromic repeats/ CRISPR-associated protein 9 (CRISPR/Cas9) into prostate cancer cells [162]. This system can deliver the CRISPR/Cas9 to bind with target gene Polo-like kinase 1 (PLK-1), which was a prosurvival gene overexpressed in the most tumors with antiapoptosis effect, resulting the significant effects of gene silence and tumor recession. Moreover, thermotherapy, which aims to cause irreversible changes in tumor cells and further lead to their apoptosis by using thermal energy at specific temperature, has become the fifth largest tumor treatment after radiotherapy, chemotherapy, surgery and immunotherapy [163]. Jo et al. developed dual aptamer-modified gold nanostars for selectively inducing apoptosis by generating considerable heat (PSMA aptamer for PSMA (+) cancer cells and DUP1 aptamer for PSMA (−) cells) [164]. When treating certain untreatable malignancies, the combination of two or more therapy methods will greatly improve the curative effect and allow for the use of lower doses to reduce the intolerable side effects [165]. Zhang et al. developed a nanodrug codelivery system, which simultaneously carried both hydrophobic DOX and hydrophilic DTX chemotherapeutic drugs and codelivered them to PSMA overexpressed cells specifically [166]. Kim et al. reported other chemo-gene combined therapeutic conjugates that PEG-grafted polyethyleneimine (PEI) was used as a vehicle for short hairpin RNAs (shRNAs) against recombinant human B-cell leukemia/lymphoma xL (Bcl-xL) delivery, and its surface was conjugated with DOX-loaded PSMA aptamer [167].

In another example, anti-PTK7 DNA aptamer sgc8 was used by Taghdisi et al. to specifically deliver daunorubicin to acute lymphoblastic leukemia T cells [168]. To enhance the stability of aptamer, Huang et al. developed an aptamer-directed anticancer drug by covalently conjugating the DOX with sgc8 aptamer through an acid-labile linker [169]. In order to increase aptamer-drug payload capacity, Zhu et al. reported another DOX delivery system named aptamer-tethered DNA nanotrains (aptNTrs) with the drug/sgc8 aptamer-NTr molar ratio of 50:1, showing the inhibition of tumor cell growth in vitro and in vivo [170]. Other nanoparticle carriers, such as mesoporous silica nanoparticles (MSN), are also an important way to improve the therapeutic efficacy. Yang et al. developed an sgc8 aptamer-modified MSN to deliver DOX to leukemia cells with high therapeutic efficacy and reducing toxicity [171]. As for the gene therapy, Zhang et al. recently developed a circle Y-shaped aptamer-DNAzyme conjugate for highly efficient in vivo gene therapy through RNA cleavage by DNAzyme [172]. This oligonucleotide drug provided a novel approach for practical therapeutic applications. In addition, Zong et al. proposed an aptamer modified composite drug nanocarrier based on black phosphorus nanosheets (BPNS) for targeted and synergetic chemophotothermal therapy of acute lymphoblastic leukemia [173]. The results demonstrated the improving curative efficiency.

The nucleolin-targeted AS1411 aptamer is also frequently used as the delivery agent. Different drug molecules can also covalently bind to AS1411 aptamer through a very simple conjugation strategy. For the first time, Trinh et al. conjugated AS1411 aptamer with DOX to targeted delivery of DOX to human liver cancer cell line (Huh7 cells) in vitro and in a murine xenograft model of hepatocellular carcinoma [174]. Other drugs, such as melittin, an amphipathic peptide derived from the honeybee venom, was also conjugated to AS1411 by Rajabnejad and coworkers [175]. In order to enhance the drug payload capacity, researchers have been using various nanocarriers, such as polymeric nanocarriers [176,177,178,179,180,181,182,183,184,185,186], graphene-based nanoparticles [187,188], liposome [189,190,191,192,193], different DNA nanostructures [194,195,196,197,198,199] and other nanoparticles [200,201]. In addition, combination therapy based on AS1411 has also been developed, Han et al. developed an AS1411 aptamer decorated immune high density lipoprotein nanostructure containing DOX for enhancing drug delivery and amplifying chemoimmuno therapy [202]. In order to further increase the therapeutic efficacy, Zhao et al. developed a ZnO-gated porphyrinic metal-organic framework (porMOF)-AS1411 nanosystem for targeted cancer therapy [203]. The pores of this nanosystem were opened by ZnO disintegration in the acidic lysosomal environment, allowing for drug release and further assembly of AS1411 aptamer to confer the targeting ability for nucleolin-overexpressed cells. This result showed highly curative efficiency without undesirable side effect.

Membrane-associated glycoforms of mucin glycoproteins, another essential class of tumor surface markers, are specifically and abundantly expressed in a variety of epithelial cancer cells [204]. Since these surface receptors can be internalized, they can also be utilized as the delivery ligand. Tan et al. developed a PEGylated anti-MUC1 aptamer-DOX complex for targeted drug delivery to breast cancer cells, demonstrating the macrophage evasion of drug could be decreased by PEGylation. To further increase the drug payload capacity, various aptamer-conjugated nanoparticles were developed as carriers for cancer therapy, such as polymeric nanocarrier [205,206], DNA nanocarriers [196,207,208,209], quantum dots and DOX [210], superparamagnetic iron oxide nanoparticles [211], chitosan nanoparticles [212,213,214], etc. In immunotherapy, Stecker et al. developed the MUC1 aptamer-C1q conjugate to recruited C9 to the cell surface, thereby increasing the formation of membrane attack complex (MAC) [215]. The results confirmed the MAC formation on the plasma membrane, leading to osmotic swelling and cell death. In radiotherapy, Borbas Ke et al. radiolabeled MUC1 aptamers using four chelators capable of forming tetrameric aptamer complexes, thereby reducing renal clearance of radioactive compounds and increasing their residence time in the blood [216]. The results indicated that the complex had better tumor penetration compared with the antibody. Azhdarzadeh et al. designed aptamer nanocomposites for magnetic resonance imaging and photothermal therapy of colon cancer cells based on photothermal therapy [217]. MUC1 aptamer was modified on superparamagnetic iron oxide nanoparticles, and the surface was coated with gold to reduce cytotoxicity and provide photothermal therapy. The results showed that the uptake of aptamer complexes was higher than that of non-targeted nanocomplexes. Kurosaki T et al. used the MUC1 aptamer for gene delivery therapy, binding the aptamer to the plasmid complex through electrostatic interactions, and the results showed that compared with the non-targeting plasmid complex, the aptamer-plasmid complex showed a more significant effect [218].

In addition to the above-mentioned cancer cell surface membrane proteins, aptamers targeting melanoma antigen (MAGE) [219], HER2 [220,221], T cell surface marker CD133 [222,223], and EpCAM [224,225,226] in vivo delivery therapy have also been developed accordingly and are summarized in the Table 6.

#### 3.2.2. Intracellular Targeted Therapy

In addition to targeting cancer surface membrane biomarkers, aptamers can also be employed in targeting intracellular biomarkers to trigger drug release. For example, the use of the “molecular currency unit” in biological energy transfer-adenosine triphosphate (ATP) as a trigger for therapeutic delivery. The stark concentration contrast between the extracellular (<0.4 mM) and intracellular (1–10 mM) environments makes ATP a viable cue for regulating drug release [228,229]. Mo et al. developed a polymeric nanogel based nanocarrier (Figure 4a), which functionalized with an ATP-binding aptamer-incorporated DNA motif, to selectively release the intercalating DOX by a conformational change when in an ATP-rich environment [229]. The results showed the half-maximal inhibitory concentration of ATP-responsive nanovehicles was 0.24 µM in MDA-MB-231 cells (human breast cancer cell line), representing a 3.6-fold increase in cytotoxicity over non-ATP responsive nanovehicles. Recently, Sun et al. designed an aptamer-based alginate hydrogel to enable ATP-responsive release of immune adjuvant Cytosine Guanine linear dinucleotide (CpG oligonucleotides) synchronized with repeated chemotherapy or radiotherapy to boost antitumor immunity [230] (Figure 4b). These ATP-responsive drug release systems provided a more complex drug delivery platform that can discriminate ATP levels to allow for drug release selection. Apart from the design directly utilizing innate intracellular ATP for drug release, a co-delivery system that employed extrinsically supplemented ATP was created using liposomes in case the intrinsic ATP level in targeted cells was insufficient to stimulate drug release [231]. In this study, directly supplying extrinsic liposomal ATP boosted drug release from the fusogenic liposome in the acidic intracellular compartments upon a pH-sensitive membrane fusion, and anticancer efficacy was increased both in vitro and in vivo. Moreover, the aptamer can also serve as a “cage” group toward nanomaterials for loading and releasing various drugs. Mo et al. developed an ATP-responsive controlled release system consisting of mesoporous silica nanoparticles (MSN) functionalized with an aptamer as cap [232]. The ON/OFF state of this system was controlled by the concentration of ATP. Aside from MSN, the ATP aptamer also served as cap for other nanomaterials (such as graphene oxide [233], CaCO_3_ [234], etc.) to achieve ATP responsive drug delivery.

At the same time, the aptamer can also be utilized for targeting subcellular organelles to realize the targeting delivery of maximal amounts of therapeutic drugs to the subcellular site in cancers. For specific nuclear targeting, Dam et al. developed a nuclear-targeted gold nanoconstruct (AuNS) functionalized with AS1411 aptamer (Apt-AuNS) [235] (Figure 4c). The nucleolin in cancer cells is overexpressed not only on the cell membrane but also on the nuclear membrane [236]. Thus, Apt-AuNS could be actively transported into cancer cells and accumulated around the nucleus, causing considerable changes in nuclear morphologies (such as nuclear envelope invaginations) at the site of AuNS nanoparticles. In order to realized mitochondria-targeted drug delivery, Ju et al. employed cytochrome c-specific binding to make mesoporous silica-coated gold nanorods for efficiently accumulating in the mitochondria of cancer cells [237] (Figure 4d). This nanostructure can load a variety of hydrophobic therapeutic drugs that acted on the mitochondria to improve curative efficacy and the near-infrared treatment can induce the release of cytochrome c and the initiation of the mitochondrial apoptosis pathway. Moreover, Zamay et al. selected an aptamer NAS-24 for targeted delivery to vimentin, which expressed on nucleus, endoplasmic reticulum or mitochondria in various human epithelial cancers [238]. They injected the mixture of arabinogalactan and NAS-24 intraperitoneal into mice with adenocarcinoma and the results exhibited cancer growth inhibition. However, due to the lack of effective organelle-targeted aptamer SELEX technologies, the organelle-targeted aptamers are relatively few to date. Even though several organelle-targeted aptamers have been reported so far, their endocytosis escape mechanisms remain to be investigated.

## 4. Conclusions and Future Prospects

To a certain extent, aptamers can be considered as chemical antibodies because they are comparatively small, chemically synthesized molecules with high specificity and affinity for almost every substance in the body, but without immunogenic responses in vivo. Meanwhile, the detection of relevant disease indicators based on the principle of the interaction of aptamers with their target biomarkers is becoming a new research field. The use of aptamers can enrich and identify the known and unknown proteins and facilitate the discovery of new cancer biomarkers. So far, aptamer-based clinical trials have yielded encouraging results, further enhancing the development of aptamer-based cancer diagnostic and therapeutic technologies. Various aptamer-targeted cancer biomarkers are differentially distributed in the cell membrane, extracellularly or intracellularly. Thus, in this review, we summarized the aptamer-based probes for cancer diagnosis and therapy based on different localization of target biomarkers. The principles of cancer diagnosis and treatment were also discussed. Although many aptamer-based probes have been designed, high-sensitive probes and their diverse applications are still worth developing.

Table 7 summarized the nucleic acid aptamers that are in various stages of development or are being approved. We can see not many aptamers in clinic, which indicates that the clinical transformation of the aptamer probes still faces many difficulties. The first issue is a scarcity of high-quality DNA aptamers for many clinically significant targets, such as glucose, cholesterol, and other cancer biomarkers. The detailed sequences information of many aptamers is not published. Second, some selected aptamers lack sufficient specificity in the face of structurally similar targets. For example, the DNA polymerase β targeted aptamer can also bind with DNA polymerase κ from another enzyme family [239], which need to introduce negative selection in the SELEX process to solve these problems [240]. Hence, a more stringent SELEX procedure is necessary to generate highly specific aptamers, facilitating their application in practical clinical diagnostics. Third, many reported aptamers lack detailed characterization, making further development difficult for analytical chemists, resulting in most studies focusing on the same model aptamers [241]. Fourth, the practical clinical diagnostic application needs to be carried out on complex samples, such as blood, serum, urine, saliva, etc. [242]. These complex samples always contain other proteins, which may influence the accuracy of diagnosis. Based on these, aptamer-based analysis requires significant testing to establish its accuracy, specificity, reproducibility, and user-friendliness, while avoiding additional steps of liquid sample transfer and mixing [242].

In cancer therapy, aptamer-based delivery systems are expected to play a critical role in the clinical treatment of cancer in the future, but many strategies are only applied in the laboratory. Most suitable ligands are only in the clinical trial stage, and only one has been approved by the US Food and Drug Administration (FDA) [243]. Meanwhile, there are currently very few screened aptamer sequences connected to diseases available. Therefore, more aptamers need to be screened and many key obstacles remain to be addressed. Firstly, aptamers are susceptive to external substances in clinical therapeutic applications, such as nonspecific serum-binding proteins in the complex internal environment of the human body, which will eventually reduce the binding efficiency of the target biomarkers. Some screened aptamers have low dissociation constants in vitro but fail to meet the target affinity in vivo [244]. Secondly, there are still some challenges in practical clinical applications because aptamers are prone to DNA degradation in the presence of nucleases in human blood. Improving the aptamers’ resistance to degradation is an effective strategy to increase its stability in human body, extend its circulation time and maintain its activity. Thirdly, there are also many shortcomings in the knowledge of pharmacokinetics, pharmacodynamics and biotoxicity of aptamer-based delivery systems. The development of aptamers with clinically effective therapeutic effects has lagged far behind that of antibodies. A number of their intrinsic physicochemical qualities need to be improved.

Aptamers’ numerous excellent properties have drawn attention. In the past decades, as research efforts are increasingly inclined toward targeted diagnosis and therapy of cancers, aptamers have also demonstrated their unique advantages and extensive application potentials. The present emphasis of attention is on how to utilize the complimentary advantages of aptamers and other materials to design more innovative, effective and rational diagnostic and delivery systems. Although there is still a long way to go, we still believe that aptamers will have a broad future with the application of interdisciplinary knowledge as the research progresses and the performance continues to improve.life-12-01937-t007_Table 7Table 7Nucleic acid aptamers currently in the clinic.Aptamer NameDNA/RNATargetTargeted DiseaseClinical StatusRef.Pegaptanib (PEGylated), VEap121RNAVEGF165Age-related macular degeneration (AMD)Diabetic macular edemaDiabetic retinopathyIn market[245]AS1411DNANucleolinAcute myeloid leukemiaMetastatic renal cell carcinomaPhase II (NCT01034410)Phase II (NCT00740441)Phase I (NCT00881244)[246]68Ga-Sgc8DNAPTK7/ CCk-4Colorectal cancerEarly Phase 1(NCT03385148)[247]ARC1905(Zimura)RNAC5Dry AMDIdiopathic polypoidal choroidal vasculopathy (IPCV)Phase I completed, Phase IIand III recruiting(NCT02686658)Zimura in Combinationwith Anti-VEGF Therapy inSubjects with IPCV(NCT02397954)[149]E-10030(Fovista)DNAPDGFNeovascular AMDPhase II (NCT02214628)Anti-PDGF PegylatedAptamer with Lucentis(NCT01089517) forneovascular AMDFovista in Combination withLucentis as compared toLucentis monotherapy(NCT01940900)[248]REG1anticoagulationsystem (RB006 andRB007)(RB006)Antidote (RB007)RNACoagulation factorIXaAcute coronarysyndromeCoronary artery diseasePercutaneous coronaryinterventionPhase I and II completed(NCT00113997,NCT00932100,NCT01872572)[249]ARC1779DNAVon Willebrandfactor (A1 domain)Purpura, ThromboticThrombocytopenic vonWillebrand diseaseAcute myocardialinfarctionPhase II (NCT00632242)Phase II (NCT00507338)[250]NU172DNAThrombinHeart diseasePhase II (NCT00808964)[251]NOX-A12RNACXCL12Chronic lymphocytic leukemiaMultiple myelomaColorectal cancerPancreatic cancerPhase II (NCT01486797)NOX-A12 in Combinationwith Bortezomib andDexamethasonePhase II (NCT01521533)[252]NOX-E36RNAMCP-1Chronic InflammatoryDiseasesType 2 diabetes MellitusSystemic LupusErythematosusPhase I (NCT00976729)[253]NOX-H94(lexaptepid pegol)RNAHuman HepcidinAnemia of chronicdiseaseEnd-stage renal diseasePhase I and II(NCT02079896)[254]ApTOLLDNAToll-like receptor (TLR4)Acute ischemic strokeAcute myocardial infarctionPhase I (NCT04742062)[255] 

## Figures and Tables

**Figure 1 life-12-01937-f001:**
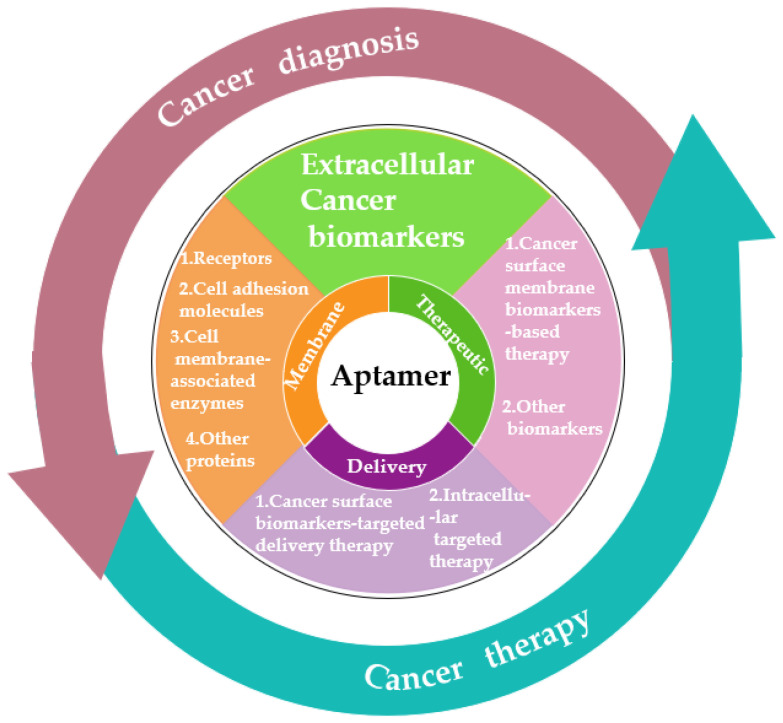
Schematic diagram of the application classification of aptamers in cancer diagnosis and treatment in the past decade.

**Figure 2 life-12-01937-f002:**
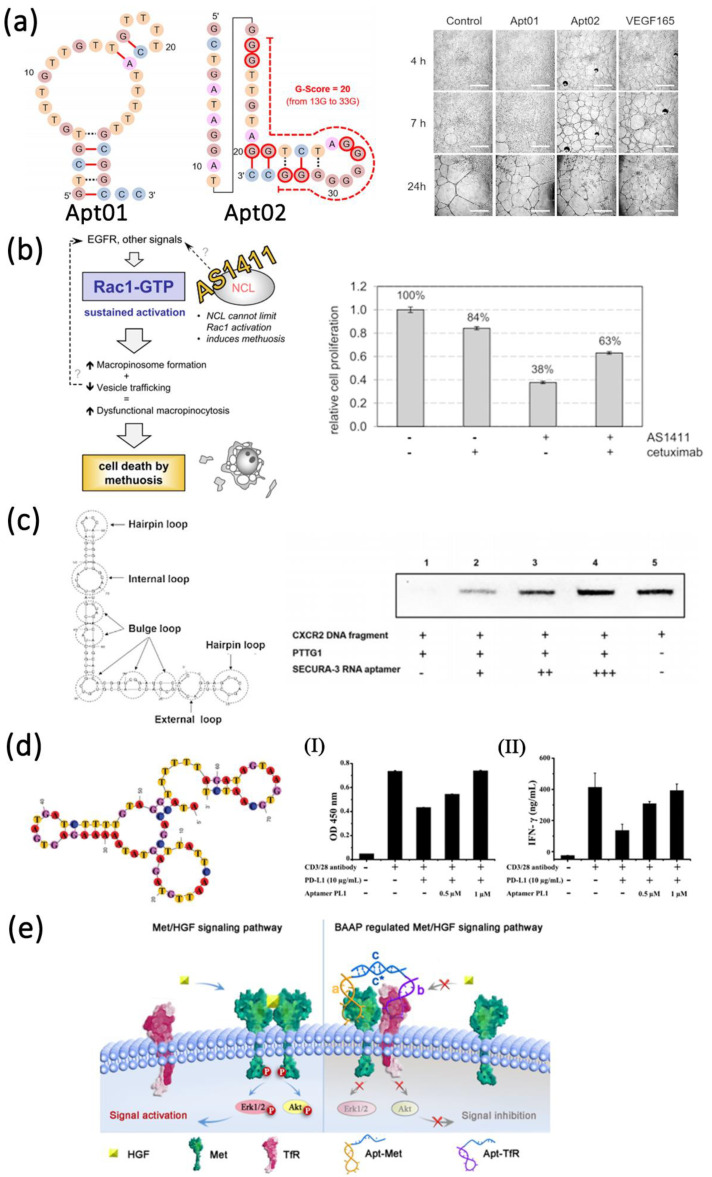
Examples of the application of aptamers as antagonists for targeted therapy of cell membrane surface biomarkers. (**a**) Schematic diagram of the structure of VEGFR-1 and VEGFR-2 aptamers and in vitro tube formation assay. Scale bars, 1 mm. Reprinted with permission from [132]. (**b**) Proposed mechanism for AS1411-induced cancer cell death and cell proliferation rate in the absence or presence of cetuximab and AS1411 aptamer. Reprinted with permission from [133]. (**c**) Secondary structure of SEURA-3 RNA aptamer and competitive binding assay. Reprinted with permission from [134]. (**d**) Secondary structure of PL1 aptamer, aptamer-mediated proliferation (**I**) and rescue of IFN-γ release (**II**) in human CD4^+^ T cells. Reprinted with permission from [136]. (**e**) Schematic diagram of bispecific aptamer targeting TfR and MET therapy. Reprinted with permission from [138].

**Figure 3 life-12-01937-f003:**
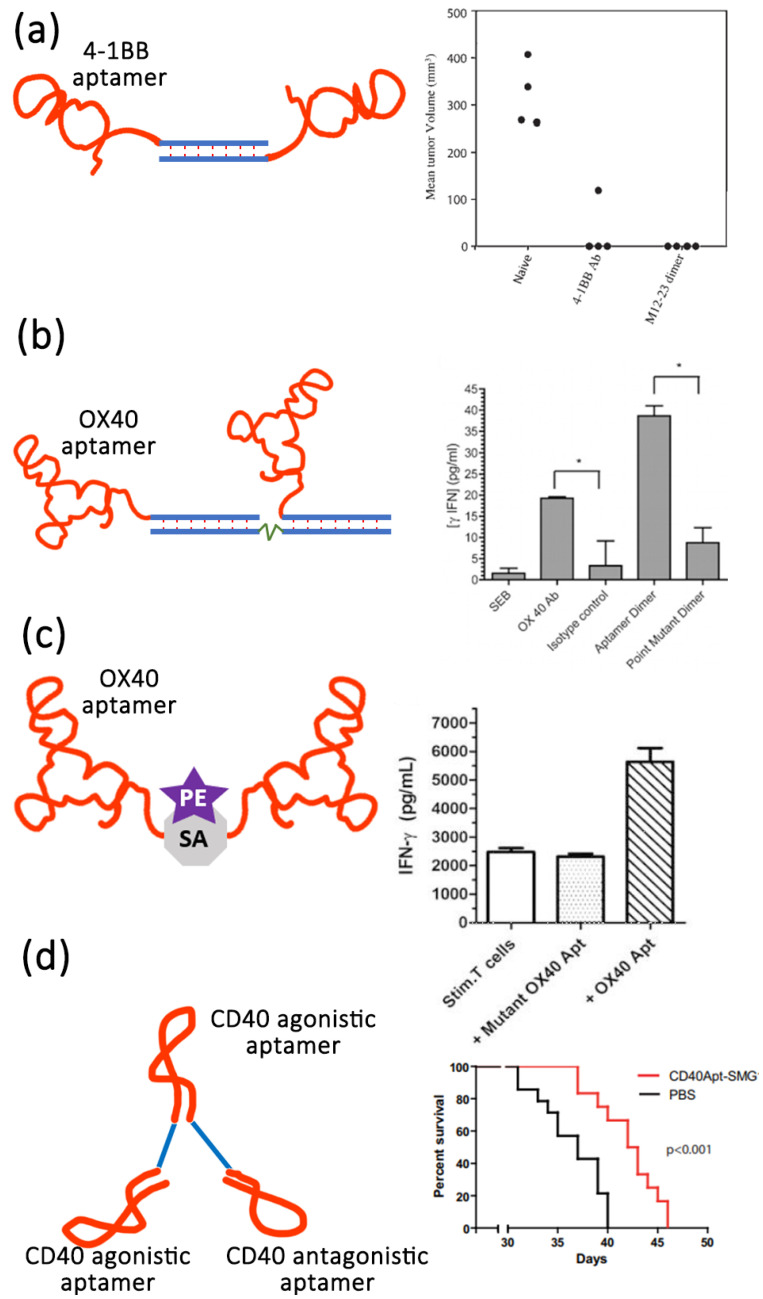
Examples of the application of aptamers as agonists for targeted therapy of cell membrane surface biomarkers. (**a**) Schematic of bivalent 4-1BB aptamer and its efficiency of tumor suppression. Reprinted with the permission from [141]. (**b**) Schematic of bivalent OX40 aptamer and OX40 activation leads to increased IFN-γ release. * *p* < 0.05. Reprinted with the permission from [148]. (**c**) Schematic of bivalent OX40 aptamer lined with streptavidin (SA)-phycoerythrin (PE) and OX40 activation leads to increased IFN-γ release. Reprinted with the permission from [142]. (**d**) Schematic of CD40Apt-SMG1-shRNA and its impact on overall survival. Reprinted with the permission from [143].

**Figure 4 life-12-01937-f004:**
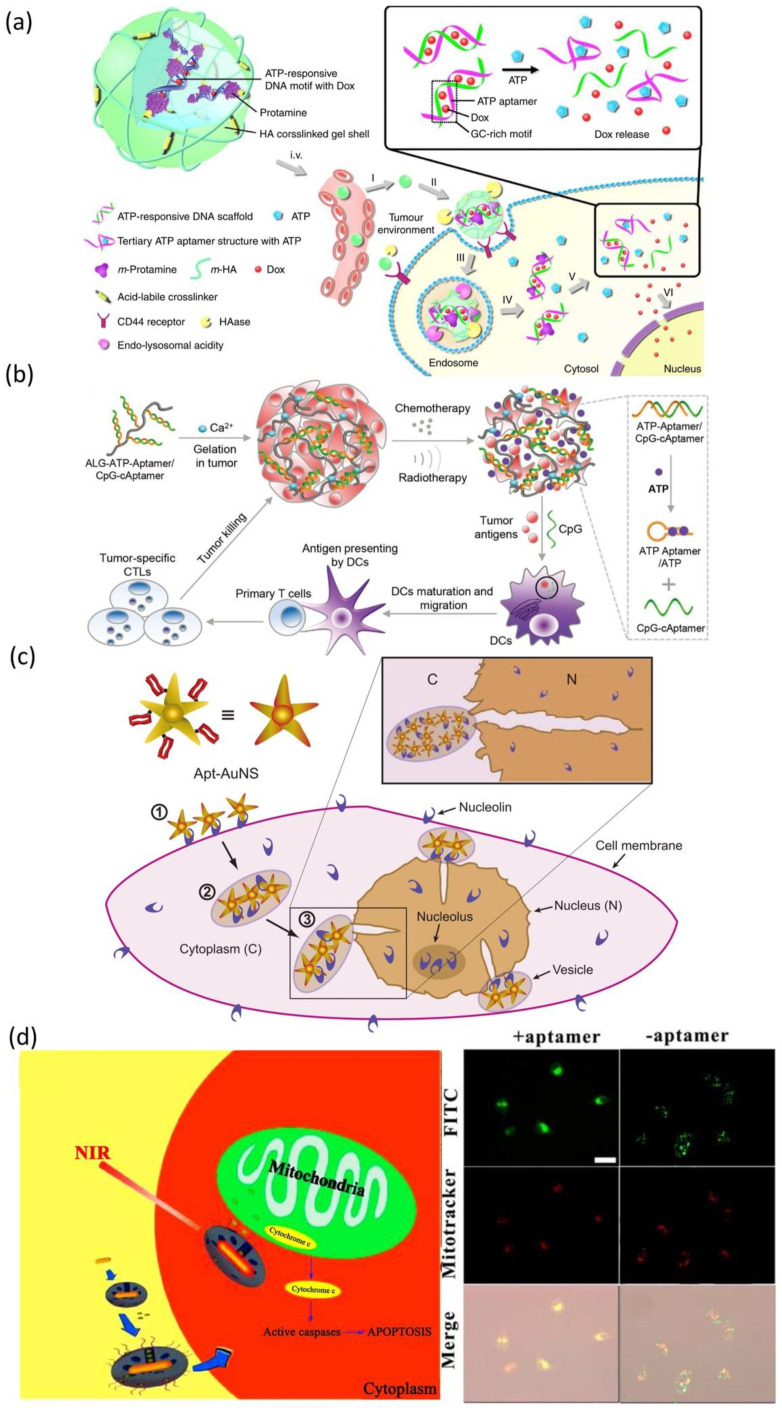
(**a**) Schematic diagram of the ATP-targeting aptamer-polymer nanogel complex and cytofluorescence image. Reprinted with permission from [229]. (**b**) Schematic diagram of the design of aptamer-binding alginate hydrogel targeting ATP and mouse fluorescence imaging image. Reprinted with permission from [230]. (**c**) Schematic diagram and cytofluorescence image of aptamer-binding AuNS nanoparticles targeting nucleolin protein on nuclear membrane. Reprinted with permission from [235]. (**d**) Design and cytofluorescence image of aptamer-binding mesoporous silica-coated gold nanorods. Reprinted with permission from [237].

**Table 1 life-12-01937-t001:** Examples of aptamers specifically targeting receptors of cell surface membrane biomarkers used in cancer diagnosis.

Biomarkers	Aptamer Name	Diagnosis Mechanism	DNA/RNA	Limit of Detection (LOD)	Cancer Type	Ref.
EGFR	Anti-EGFR aptamer	Aptamer/antibody based immunosensor.	DNA	50 pg/mL	Human epidermal squamous cell carcinoma	[22]
	Anti-EGFR aptamer	Origami-paper-based graphene-modified aptasensor.	DNA	5 pg/mL	Nonsmall-cell lung cancer	[23]
	MinE07	^18^F-labeled RNA aptamer for PET imaging.	RNA	-	Lung cancer	[24]
VEGFR2	Anti-VEGFR2 aptamer	Aptamers-modified magnetic nanoparticles for MR imaging	DNA	-	Breast cancer	[27]
HER2	HB5	Proximity-induced fluorescence activation of aptamers and G-rich sequences templated AgNCs.	DNA	0.0904 fM	Breast cancer	[28]
		LC-MS/MS-based quasi-targeted proteomics strategy coupled with aptamers-triggered hybridization chain reactions.	DNA	-	Breast cancer	[29]
		Proximity-induced fluorescence activation of aptamer-templated AgNCs.	DNA	220 pM	Breast cancer	[32]
PTK7	Sgc8	^18^F labeled aptamer for PET imaging.	DNA	-	Malignant melanoma	[33]
		Electrochemical aptamer-based determination by toehold-mediated strand displacement amplification on AuNPs and GOs.	DNA	1.8 fM	Colon cancer	[34]
CD4	Anti-CD4 aptamer	Aptamers containing surfaces for capture of CD4 expressing cells.	RNA	-	HIV/AIDS	[37]
CD30	CD30	Aptamer conjugated IRD800CW reporter for imaging.	RNA	-	Lymphoma	[39]
		Cyanine dye (Cy5) labeled aptamers for flow cytometry detection of CD30 -expressing cells.	RNA	0.3 nM	Lymphoma	[40]
CD63	CD63	Aptamer accelerated intrinsic peroxidase-like activity of g-C3N4 nanosheets for detection of exosomes.	DNA	13.52 × 10^5^ particles/μL	Breast cancer	[41]
		Aptamer capped single-wall carbon nanotubes based colorimetric aptasensor for detection of exosomes.	DNA	5.2 × 10^2^ particles/μL	Breast cancer	[42]
		Aptamer and G-quadruplex/hemin DNAzyme modified ECL sensor for detection of exoxomes.	DNA	7.41 × 10^4^ particles/mL	Breast cancer	[43]
		Aptamer linked dual AuNPs based sensor for detection of exosomes by surface plasmon resonance.	DNA	5 × 10^3^ particles/mL	Breast cancer	[44]
		Aptamer-based fluorescence polarization for separation of free exosomes quantification.	DNA	500 particles/μL	Breast cancer	[45]

**Table 2 life-12-01937-t002:** Examples of aptamers specifically targeting CAMs of cell surface membrane biomarkers used in cancer diagnosis.

Biomarkers	Aptamer Name	Diagnosis Mechanism	DNA/RNA	LOD	Cancer Type	Ref.
EpCAM	EpCAM	Aptamer-based graphene quantum dots and MoS_2_ nanosheets for detection of target through FRET.	DNA	450 pM	Breast cancer	[48]
		Enzyme-free fluorescence detection of EpCAM by a combined aptamer-based recognition and toehold-aided DNA recycling amplification strategy.	DNA	0.1 ng/mL	Breast cancer	[49]
		Aptamer conjugated maleimidyl magnetic nanoplatform for facile MRI.	DNA	-	Breast cancer	[50]
		Graphene oxide and fluorescent labeled aptamers for detection of exosomes.	DNA	2.1 × 10^4^ particles/µL	Colorectal cancer	[51]
		Aptamer modified Ti3C2 MXenes nanosheets based chemiluminescence biosensor for detection of exosomes.	DNA	125 particles/µL	Breast cancer	[53]
		EpCAM and CD63 aptamer-based 3D DNA walker amplification and Exo III-assisted electrochemical ratiometric detection of exosomes.	DNA	1.3 × 10^4^ particles/mL	Breast cancer	[55]
		Aptamer-based gold nanostars for detection of EpCAM overexpressed-CTCs by RCA coupled with the hemin/G-quadruplex complex.	DNA	1 cell/mL	Breast cancer	[56]
Integrins	Apt_αvβ3_	Aptamer conjugated magnetic nanoparticles for detection of integrin ανβ3 by MRI.	DNA	-	Glioblastoma	[57]
	H02	Aptamer as probes for cyto- and histofluorescence.	RNA	-	Glioblastoma	[58]

**Table 3 life-12-01937-t003:** Examples of aptamers specifically targeting cell membrane-associated enzymes used in cancer diagnosis.

Biomarkers	Aptamer Name	Diagnosis Mechanism	DNA/RNA	LOD	Cancer Type	Ref.
PSMA	xPSM-A9 xPSM-A10	Aptamer as probes and processed the ability to inhibit N-acetyl-α-inked acid dipeptidase.	RNA	2.1 nM 11.9 nM	Prostatic cancer	[61]
	xPSM-A10	Aptamer conjugated AuNPs for detection of target by utilizing tissue microarrays.	RNA	-	Prostatic cancer	[62]
	A10-3.2	Aptamer-oriented lipid nanobubbles as ultrasound contrast agent.	RNA	-	Prostatic cancer	[63]
	A10	Aptamer conjugated multi-walled carbon nanotubes as ultrasound contrast agent.	RNA	-	Prostatic cancer	[64]
		RNA/peptide dual-aptamer probe based electrochemical detection.	RNA	-	Prostatic cancer	[66]
MMP-9	F3B	^99m^Tc labeled aptamer for ex vivo imaging slices of human brain tumors.	RNA	20 nM	Glioblastoma	[68]
		Dual aptamer-based piezoelectric biosensor for the detection of target.	RNA	1.2 pM	-	[69]
		Programmed hybridization/dehybridization of aptamers on the surface of AuNSs based photoacoustic contrast agent.	RNA	-	Breast cancer	[70]

**Table 4 life-12-01937-t004:** Examples of aptamers specifically targeting other membrane associated proteins used in cancer diagnosis.

Biomarkers	Aptamer Name	Diagnosis Mechanism	DNA/RNA	LOD	Cancer Type	Ref.
MUC1	MUC1	Fluorescence aggregation assay by carbon dot-labeled antibodies and aptamers.	DNA	2 nM	-	[74]
		Aptamer-tagged AgNCs for fluorescent imaging.	DNA	0.05 nM	Breast cancer	[75]
		Four-way branch migration-based strategy for amplified aptamer detection of MUC1.	DNA	2.8 nM	-	[76]
		Dual-aptamer (VEGF and MUC1 aptamers) nanoparticle-mediated signal amplification strategy for cancer cells colorimetric detection.	DNA	10 cells/mL	Breast cancer	[77]
		Aptamer conjugated carbon nanospheres for electrochemical detection of target.	DNA	40 cells/mL	Colon cancer	[78]
		Label free aptasensor for electrochemical detection of target by combining hemin/G-quadruplex system and RCA.	DNA	9.54 × 10^2^ particles/mL	Gastric cancer	[80]
Nucleolin	AS1411	Aptamer-based microcantilever biosensor for the detection of target.	DNA	1.0 nM	-	[82]
		^64^Cu labeled aptamer for tumor-targeted imaging by microPET/CT.	DNA	-	Lung cancer	[83]
		Aptamer conjugated with HYNIC and ^99m^Tc for thin layer chromatography.	DNA	-	Prostate cancer	[84]
		Cobalt-ferrite nanoparticle surrounded by fluorescent rhodamine within silica shell matrix conjugated with aptamer for multimodal cancer-targeted imaging.	DNA	-	Glioma	[85]
		AS1411 aptamer-based phosphorescent nanoprobe for tumor imaging.	DNA	-	Breast cancer	[86]
		Dual aptamer (AS1411 and CD63 aptamer) recognition-based G-quadruplex nanowires for the detection of exosomes.	DNA	1.85 × 10^3^ particles/mL	Cervical carcinoma	[87]
		MUC1 aptamer functionalized magnetic beads and AS1411 modified quantum dots based nano-bio-probes for the multimode detection of target.	DNA	201 cells/mL 85 cells/mL	Breast cancer	[88]
		Sandwich-type cytosensor based on MOF and DNA tetrahedron linked dual aptamer (AS1411 and MUC1) for electrochemical detection of target.	DNA	6 cells/mL	Breast cancer	[89]
VEGF165	VEGF165	Bivalent aptamer-Cu nanocluster for fluorescent detection of target.	DNA	12 pM	Colorectal cancer	[92]
		Nicking endonuclease-assisted signal amplification of a split aptamer beacon for detection of target.	DNA	1 pM	Breast cancer	[93]
		AgNPs-enhanced time resolved fluorescence sensor for VEGF165 detection by using long-lived fluorescent Mn-doped ZnS QDs.	DNA	-	-	[94]
		Aptamer conjugated USPIO nanoparticles for MRI imaging of VEGF165-expressing tumors in vivo.	DNA	-	Liver cancer	[95]

**Table 5 life-12-01937-t005:** Examples of aptamers specifically targeting extracellular biomarkers used in cancer diagnosis.

Biomarkers	Aptamer Name	Diagnosis Mechanism	DNA/RNA	LOD	Cancer Type	Ref.
PDGF-BB	PDGF-BB	Aptamer modified AuNPs for colorimetric detection of target.	DNA	32 nM	-	[100]
		Aptamer-based DNA enzyme-catalyzed colorimetric reaction coupled with RCA for colorimetric detection of target.	DNA	8.2 fM	Breast cancer	[101]
		AuNPs labeled and target-triggered strand displacement amplification for colorimetric detection of target.	DNA	1.1 nM	Breast cancer	[102]
		Aptamer-based cascade amplification SERS method for the detection of target.	DNA	0.42 pM	-	[103]
		Aptamer conjugated hydrophobic Ru (II) complex for label-free luminescent detection of target.	DNA	0.8 pM	-	[104]
		Target-triggered hybridization chain reaction amplification and GO-based selective fluorescence quenching.	DNA	1.25 pM	-	[105]
		Photo-induced electron transfer between aptamer-AgNCs and G-quadruplex/hemin complexes.	DNA	1 × 10^−13^ M	-	[106]
		FRET based aptasensor using UCNPs as donor and AuNPs as acceptor for the detection of target.	DNA	10 nM	Lymphoma	[107]
		The precipitates’ electrochemical aptasensing between the reaction of phosphate group in both HAP-NPs and the aptamer reacted with molybdate.	DNA	50 fg/mL	-	[110]
AFP	AFP	Target-induced aptamer switched mode for label-free fluorescent detection of target.	DNA	1.76 nM	-	[115]
		Sandwich binding type based fluorescent aptasensor for detection of target.	DNA	400 pM	Hepatocellular carcinoma	[116]
CEA	CEA	Fluorophore-labeled aptamer-absorbed MoS_2_ nanosheets for detection of target.	DNA	34 pg/mL	Gastrointestinal neoplasms	[118]
		Aptamer linking AgNCs with AuNPs for fluorescent detection of target.	DNA	3 pg/mL	-	[119]
		Aptamer-based FRET sensor between NIR-CDs and AuNRs for fluorescent detection of target.	DNA	0.02 pg/mL	Lung cancer	[121]
		Aptamer-based dsDNA templated copper nanoparticles for label-free fluorescent detection of target.	DNA	6.5 pg/mL	-	[122]
		Exonuclease III-assisted target recycling and DNA walker amplification strategy for fluorescent detection of target.	DNA	1.2 pg/mL	-	[123]
8-OHdG	8-OHdG	Aptamer combined with NMM fluorophore to form a fluorescent switch to detect target.	DNA	1.19 nM	-	[126]
		Aptamer-based 3D DNA nanomachine for fluorescent detection of target.	DNA	4 pM	-	[127]
		Target-triggered polyaniline deposition on aptamer-based tetrahedral DNA nanostructure for electrochemical detection of target.	DNA	1 pM	Bladder cancer	[128]
		Fe_3_O_4_-aptamer magnetic nanoparticles for the detection of target by HPLC-MS.	DNA	0.01 nM	-	[129]

**Table 6 life-12-01937-t006:** Example of aptamers specifically targeting cancer surface biomarkers used in cancer therapy.

Biomarkers	Aptamer Name	Delivery Mechanism	DNA/RNA	Cancer Type	Ref.
PSMA	A10	Dtxl-encapsulated nanoparticles formulated with PLGA-b-PEG copolymer and surface functionalized with aptamers.	RNA	Prostate cancer	[158]
		DNA-RNA hybrid aptamer coupled SPION to delivery DOX.	RNA	Prostate cancer	[159]
		Aptamers anchored nanoparticles to encapsulate docetaxel for tumor delivery	RNA	Prostate cancer	[160]
		Aptamers-siRNA chimera for in vivo delivery for targeted gene silencing therapy	RNA	Prostate cancer	[161]
		Aptamer-conjugated cationic liposomes to deliver therapeutic CRISPR/Cas9	RNA	Prostate cancer	[162]
		Dual aptamer-modified gold nanostars for generating heat to induce apoptosis.	RNA	Prostate cancer	[164]
		DOX carried aptamer conjugated with DTX loaded PLGA-b-PEG nanoparticles for targeted chemotherapy.	RNA	Prostate cancer	[166]
		DOX carried aptamer conjugated with shRNA loaded PEG-PEI polyplexes for targeted chemogene therapy.	RNA	Prostate cancer	[167]
PTK7	Sgc8	Daunorubicin-carried aptamer for targeted chemotherapy.	DNA	Lymphoblastic leukemia	[168]
		DOX-carried aptamer for targeted chemotherapy.	DNA	Lymphoblastic leukemia	[169]
		DOX-loaded aptamer-tethered DNA nanotrains for targeted chemotherapy.	DNA	Lymphoblastic leukemia	[170]
		Aptamers modified MSN to deliver DOX to tumors.	DNA	Lymphoblastic leukemia	[171]
		Aptamer-DNAzyme conjugate for targeted gene therapy.	DNA	Cervical carcinoma	[172]
		Aptamer modified composite drug nanocarrier based on black phosphorus nanosheets for targeted chemophotothermal therapy.	DNA	Lymphoblastic leukemia	[173]
Nucleolin	AS1411	DOX-carried aptamer for targeted chemotherapy.	DNA	Hepatocellular carcinoma	[174]
		Melittin conjugated aptamer for targeted chemotherapy.	DNA	Human non-small cell lung cancer	[175]
		Dual aptamer modified dendrigraft poly-L-lysin nanoparticles for targeted delivery of DOX.	DNA	Cervical carcinoma	[186]
		Aptamer-decorated dextran coated nano-graphene oxide for targeted dextran delivery.	DNA	Breast cancer	[187]
		Aptamer-functionalized liposomes loaded with DOX for in vitro and in vivo targeted delivery.	DNA	Breast cancer	[189]
		Dual targeting DNA tetrahedron nanocarrier (MUC1-Td-AS1411) for targeted delivery of DOX.	DNA	Breast cancer	[194]
		Anti-miR-155-loaded MSNs modified with polymerized dopamine and aptamer for targeted gene therapy.	DNA	Rectal cancer	[200]
		Lipophilic aptamer-CpG fused sequences conjugated lipoprotein for targeted delivery of DOX.	DNA	Lung cancer	[202]
		Aptamer modified ZnO-gated porMOF-based drug delivery system for targeted delivery for targeted bimodal cancer therapy.	DNA	Cervical carcinoma	[203]
MUC1	MUC1	PEGylated aptamer-DOX complex for targeted chemotherapy.	DNA	Breast cancer	[204]
		Aptamer modified and vinorelbine loaded lipid-polymer hybrid nanoparticles for targeted chemotherapy.	DNA	Breast cancer	[206]
		Aptamer-conjugated DNA icosahedral nanoparticles as a DOX carrier for targeted chemotherapy.	DNA	Epithelial cell carcinoma	[207]
		Quantum dots-aptamer -DOX conjugates for targeted chemotherapy.	RNA	Ovarian cancer	[210]
		Epirubicin loaded super paramagnetic iron oxide nanoparticle-aptamer bioconjugate for targeted chemotherapy.	DNA	Colorectal cancer	[211]
		Aptamer conjugated chitosan nanoparticles as an anticancer SN38 carrier for targeted chemotherapy.	DNA	Colon cancer	[212]
		Aptamer-C1q protein conjugates for targeted immunotherapy.	DNA	Breast cancer	[215]
		Mono- and multimeric targeted radiopharmaceuticals based on cyclen ligands coupled with aptamer for targeted radiotherapy.	DNA	Breast cancer	[216]
		Aptamer conjugated gold coated superparamagnetic iron oxide nanoparticles for targeted photothermal therapy.	DNA	Colon cancer	[227]
		Aptamer coated pDNA/PEI complexes for targeted gene therapy.	DNA	Lung cancer	[218]
MAGE	Ap52	Phosphonothioate-modified aptamer complexed with DOX for targeted chemotherapy.	DNA	Pancreatic cancer	[219]
HER2	HB5	Aptamer conjugated mesoporous silica nanocarrier-based DOX delivery system for targeted chemotherapy.	DNA	Breast cancer	[221]
		Aptamer functionalized curcumin-loaded human serum albumin nanoparticles for targeted chemotherapy.	DNA	Breast cancer	[220]
CD133	CD133	DOX-loaded aptamer for targeted chemotherapy.	DNA	Hepatocellular carcinoma	[222]
		Aptamer-modified docetaxel liposome for targeted chemotherapy.	DNA	Lung cancer	[223]
EpCAM	EpCAM	Aptamer-functionalized mesoporous silica nanoparticles as a DOX carrier for targeted chemotherapy.	DNA	Colon cancer	[224]
		Aptamer and quantum dots-functionalized nutlin-3a loaded poly (lactide-co-glycolide) nanoparticles for targeted chemotherapy.	DNA	Breast cancer	[225]
		Aptamer functionalized PLGA-lecithin-curcumin-PEG nanoparticles for targeted chemotherapy.	DNA	Colorectal cancer	[226]

## Data Availability

Not applicable.

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
