# Peer review of "Aptamer-Based Probes for Cancer Diagnostics and Treatment"

_life, 2022, doi:10.3390/life12111937_

Round 1
Reviewer 1 Report
No Major comments
Minor comments:
1. Figure 2,3,4,6- poor image quality. Low resolution images
2. English language revision
Reviewer 2 Report
1. Language of manuscript is not good need major corrections and english editing from professional services.
2. Better to cite and mention adapted figures.
3. Scheme could be prepared for overall theme of the review.
Reviewer 3 Report
The manuscript submitted by Hu et al. deals with applications of Aptamers-Based Probes in cancer diagnostics and treatments. They have summarized the applications based on the different localization of target biomarkers, as well as the current challenges and future prospects.
1. It would be better if the author summarized the approved and clinical trial aptamers-based probes in the form of a table.
2. The review flow is monotonous. Authors are encouraged to include schematic representations to reach a wider audience.
3. A brief introduction about aptamers selection should be included
4. The authors summarized aptamer-based probes for cancer diagnosis as 2.1. Cancer surface membrane biomarkers, 2.2. Extracellular cancer biomarkers, 2.3. Cancer exosomes, 2.4. Circulating tumor cells. So, Page 5, lines 194 – 199 should be moved to 2.4. Circulating tumor cells. Similarly, in Table 1, information for reference numbers 66 and 68 may be moved to Table 3.
5. Inside the manuscript and the tables: the name of the aptamers should be mentioned clearly as indicated in the original research papers to avoid confusion.
6. Table 3. HL-60, HepG2, Ramos, and SK-BR-3 are not the name of the target. Please check.
7. Section 2.3 Cancer exosomes: Require an additional Table to consolidate the examples of applications of aptamers targeting transmembrane protein exosomes.
8. In Table 1, include the integrins as one of the transmembrane proteins.
9. Table 1. NCL and Table 5 Nucleolin protein. Please mention uniformly, NCL or Nucleolin protein.
10. This reviewer is unable to understand the difference between 2.1.3 Cell membrane-associated enzymes and 2.1.4 Other membrane-associated proteins and why this classification is required.
11. The first time you use an abbreviation, it's important to spell out the full term and put the abbreviation in parentheses. Thereafter, the author can stick to using the abbreviations. A list of abbreviations can also be supplied along with the manuscript. For example, Page 4, Line 138, AgNC?
12. Carcinoembryonic antigen (CEA) targeting aptamers were discussed in Table 1 and Table 2 separately. What is the reason behind it? Is it possible to bring them together at one Table?
13. The grammar and English style of the manuscript requires careful revision. There are numerous errors. Page 3, Line 112; Page 28, Line 877.
